# Understanding Prompt Tuning and In-Context Learning via Meta-Learning

**Tim Genewein**\*           **Li Kevin Wenliang** \*           **Jordi Grau-Moya**

**Anian Ruoss**           **Laurent Orseau**           **Marcus Hutter**

Google DeepMind

## Abstract

Prompting is one of the main ways to adapt a pretrained model to target tasks. Besides manually constructing prompts, many prompt optimization methods have been proposed in the literature. Method development is mainly empirically driven, with less emphasis on a conceptual understanding of prompting. In this paper we discuss how optimal prompting can be understood through a Bayesian view, which also implies some fundamental limitations of prompting that can only be overcome by tuning weights. The paper explains in detail how meta-trained neural networks behave as Bayesian predictors over the pretraining distribution, whose hallmark feature is rapid in-context adaptation. Optimal prompting can be studied formally as conditioning these Bayesian predictors, yielding criteria for target tasks where optimal prompting is and is not possible. We support the theory with educational experiments on LSTMs and Transformers, where we compare different versions of prefix-tuning and different weight-tuning methods. We also confirm that soft prefixes, which are sequences of real-valued vectors outside the token alphabet, can lead to very effective prompts for trained and even untrained networks by manipulating activations in ways that are not achievable by hard tokens. This adds an important mechanistic aspect beyond the conceptual Bayesian theory.

## 1   Introduction

Perhaps the most impressive feature of today's frontier models is their ability to swiftly adapt their behavior to a wide range of contexts. Given relatively few tokens—whether from a user input, a system prompt, or a number of in-context examples—models often rapidly infer the task at hand and produce good continuations without any weight adaptation (in-context learning, Lampinen et al. [2024]). From a meta-learning perspective, rapid in-context adaptation is expected to arise: log loss minimization with a parametric sequential predictor (like a neural network) over a distribution of stochastic data generators leads to a Bayesian predictor for the pretraining distribution [Ortega et al., 2019]. The hallmark feature of such a predictor (Bayes-optimality) is most rapid in-context adaptation and least (cumulative) prediction error on average. Accordingly, prompting, that is conditioning of the Bayesian predictor, can be used to data-efficiently adapt the pretrained model to a target task. An important question is: under which conditions is it possible to find a prompt such that the prompted pretrained predictor becomes (near-) Bayes-optimal on a target task? We refer to this as *optimal prompting*, which is possible in theory if the target task is one of the tasks covered by the meta-distribution. If this is not the case, then optimal prompting may not be possible for an ideal predictor, and weight adaptation may be necessary.

---

\*Correspondence to `{timgen, kevinliw}@google.com`.

39th Conference on Neural Information Processing Systems (NeurIPS 2025).

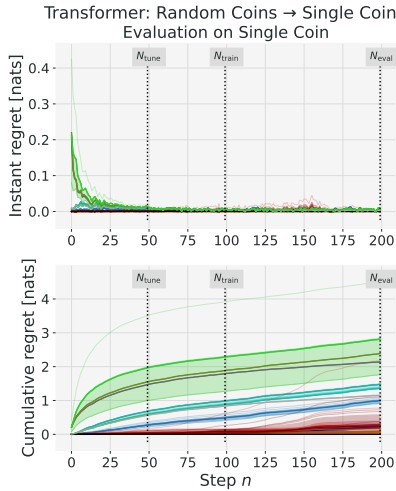

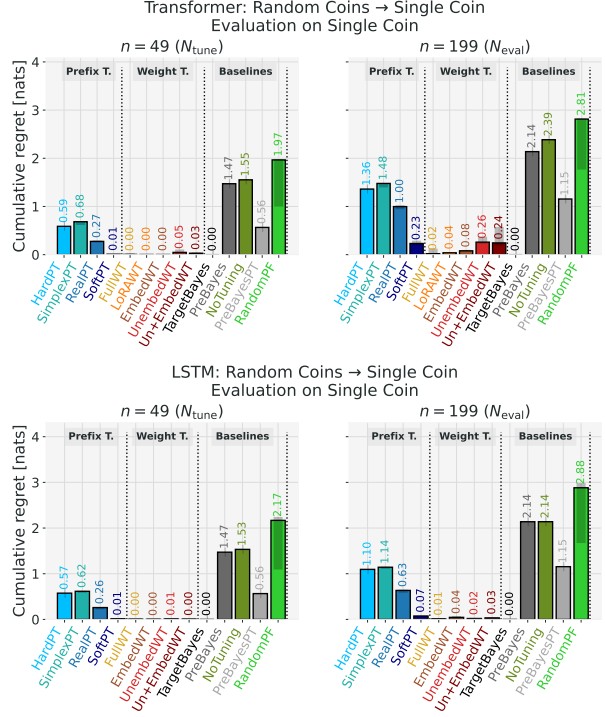

**Top:** Performance of different tuning methods on Transformers, measured as excess log loss, i.e., regret (Eq. (10), lower is better). See bar plots for color legend. **Top-right:** Detailed Transformer results for last step within the tuning sequence length $N_{\text{tune}}$ and the last evaluation step $N_{\text{eval}}$. **Right:** Like above but for LSTM.

Figure 1: Pretraining on sequences from coins with uniform random bias (length $N_{\text{train}} = 100$), then fine-tuning to the target task of a single coin with bias 0.2 (tuning sequence length $N_{\text{tune}} = 50$). Plots show prediction performance on the target task for different prefix- and weight-tuning methods. For both Transformers and LSTMs Soft Prompting ('SoftPT') leads to optimal performance, showing that networks can be successfully prompted to behave Bayes-optimally on the target distribution ('TargetBayes'). This holds up to the tuning sequence length (50), with only minor degradations up to 200 steps. The corresponding soft prefixes of length= 6 outperform even the best hard token prefixes of the same length ('HardPF'). Most weight-tuning methods also perform very well. See Section 4 for method details. Thick lines and bars show the median over 10 tuning repetitions, thin lines individual repetitions, and shaded areas/bars show $25\%, 75\%$ quantiles. See Fig. 2 for a visualization of models' internal dynamics. Regret curves for the LSTM, similar to top-left panel, are shown in Fig. A5.

The goal of prompt tuning for a target task is to produce prompts that, when consumed by the model, inject maximal information (up to statistical sufficiency) about the target task into the predictor's internal state. In practice, prompt optimization is often done by tuning soft prefixes, which are sequences of real-valued input vectors instead of hard tokens. As we show, the corresponding off-distribution inputs can exploit neural circuitry to inject substantially more information compared to even the best hard token sequence of the same length, without breaking subsequent internal dynamics.

In this paper, we investigate prefix-tuning, where a short prompt prefix is tuned to maximize subsequent prediction accuracy. We consider prefix search over hard tokens, and three soft prefix-tuning methods, including 'Soft Prompting', [Lester et al., 2021]. The code to reproduce all our experiments is available at: `https://github.com/google-deepmind/thunnini`. Our main contributions are:

- We discuss how prompting can be understood as steering a Bayesian sequential predictor via its pretrained in-context adaptation mechanism that arises from meta-training, see Section 2.

- We analyze theoretical conditions for the relationship between pretraining distribution and target task under which optimal prompting is and is not possible, see Section 3.

- We confirm these theoretical conditions empirically via a series of educational experiments, and show that in the negative case weight-based fine-tuning can succeed. See Section 4.2.

- We investigate mechanistic aspects of prompting LSTMs and Transformers, with experiments on both pretrained and untrained networks. In all cases, soft prefixes can be much

more effective than any hard token sequence of equal length, particularly for Transformers, where soft prefixes can even cause untrained networks to behave as well-performing sequential predictors. See Section 4.2.

**Limitations and Scope.** As with any conceptual principle, the Bayesian view cannot exhaustively describe all in-context learning phenomena at (frontier model) scale: limited data and model expressivity, suboptimal optimization, out-of-distribution generalization, and other mechanistic aspects will have additional impact. Having said that, there are a number of recent investigations at LLM scale supporting the plausibility of the Bayesian view, such as Chan et al. [2022a] who argue that distributional properties of natural language can easily give rise to a meta-learning setting. This paper discusses fundamental properties of prefix tuning, which we illustrate with educational experiments where the focus is on clarity and being able to compare quantitatively against a known and tractable Bayesian predictor. Accordingly, our datasets do not capture the full complexities of, e.g, large-scale language, vision, or robotics tasks. Similarly, our neural networks are small by modern standards, which means that our findings must be very cautiously extrapolated towards modern frontier model scale. Further rigorous and well-designed scientific studies are necessary to bridge the gap between our current work and modern large-scale ML practice, and we are optimistic that our fundamental results will inspire the design of such studies. The simplicity of our experimental setup allows to carefully control the data statistics, and compare neural predictors against exact Bayes-optimal predictors—both of which is not true for LLM-scale experiments. Additionally, it allows us to focus on fundamental aspects of prompting that arise even in idealized settings, and thus hold at any scale. We believe that this lays important fundamental groundwork, which will help future research to isolate and more effectively investigate additional, non-idealized, aspects of in-context learning and prompt tuning at frontier scale. We do not propose improved prompt tuning methods in this work since our aim is to study and understand existing ones.

## 2 Background: Memory-Based Meta-Learning

In this section we review how memory-based meta learning leads to Bayesian sequential predictors, whose hallmark feature is (most) rapid in-context adaptation. Under this view, the role of prompt tuning is to facilitate the inference process of the target task. While LLMs and frontier models are not explicitly meta-trained, their training process can be viewed as implicit meta-training, in which case the Bayesian view constitutes an important conceptual principle for understanding pretrained models and their in-context learning abilities.

**Tasks.** Let an alphabet $\mathcal{A}$ be a finite set of tokens (with one-hot encoding) and $x_{1:N} \in \mathcal{A}^N$ be a sequence of such tokens of length $N$, and $x_0 = \epsilon$ be the empty sequence. A task is a distribution over finite-length sequences:
$$P(x_{1:N}|\tau) : \mathbb{R}^M \to \Delta\mathcal{A}^N$$
where $\tau \in \mathbb{R}^M$ is the $M$-dimensional parameter vector of the task, over which a distribution $p(\tau)$ can be placed (the task- or meta-distribution). For notational simplicity we assume all sequences have the same length $N$. Simple examples of sets of tasks would be the family of Bernoulli distributions parameterized by the bias $\tau \in [0, 1]$, or the family of Markov processes over sequences in $\mathcal{A}^N$ parameterized by the transition kernel and initial state distribution. The set of tasks and the distribution over tasks define the marginal distribution over sequences:

$$\xi(x_{1:N}) = \int P(x_{1:N}|\tau)P(\tau)d\tau = \int P(x_N|x_{<N}, \tau)P(\tau|x_{<N})P(x_{<N})d\tau \qquad (1)$$

$$\forall n \in \mathbb{N}^+ : \quad \xi(x_n|x_{<n}) = \int P(x_n|x_{<n}, \tau)P(\tau|x_{<n})d\tau \qquad (2)$$

which we have rewritten in the second line in its "next-token predictor" form (conditional distribution over next token, given full history).

**Sequential predictor.** Let $\pi_\theta$ be a parametric sequential predictor, such as a neural network, which is a function with parameters $\theta$ that takes in an arbitrary-length sequence of $D$-dimensional float vectors and outputs a discrete distribution over the next (one-hot) token: $\pi_\theta : \{\mathbb{R}^D\}^* \to \Delta\mathcal{A}$ Typically, $D = |\mathcal{A}|$, meaning that one-hot tokens can be fed directly into the predictor. The predictor's

conditional distribution over the next token given a context $x_{<n}$ is given by evaluating the predictor with the given context (i.e., performing a forward pass): $P_\theta(x_n|x_{<n}) = \pi_\theta(x_{<n})$. Keep in mind that a neural net accepts not only hard tokens as inputs, but sequences of arbitrary vectors in $\mathbb{R}^D$ ('soft tokens'), and will produce a distribution over the next hard token in either case.

**Meta-Learning.** In memory-based meta learning, $\theta$ is adjusted by repeating the following steps:

1. *Reset memory:* set predictor's internal state or context to a fixed initial value.
2. *Sample task:* $\tau \sim P(\tau)$.
3. *Generate data:* sample one or more sequences from the task: $x_{1:N} \sim P(x_{1:N}|\tau)$.
4. *Update parameters:* perform a gradient step towards minimizing prediction error (log loss) on the sampled sequences.

Log loss is the cumulative prediction error. For a single sequence it is:

$$\mathscr{L}_\theta(x_{1:N}) := -\log \pi_\theta(x_{1:N}) = -\sum_{n=1}^{N} \log \pi_\theta(x_n|x_{<n}) \tag{3}$$

The expected *excess log loss* measures how much worse $\pi_\theta$ performs w.r.t. expected cumulative prediction error compared to the best possible predictor that does not know $\tau$:

$$\mathbb{E}_\xi\left[-\log \pi_\theta(x_{1:N})\right] - \mathbb{E}_\xi\left[-\log \xi(x_{1:N})\right] = D_{\mathrm{KL}}(\xi||\pi_\theta) \geq 0 \tag{4}$$

which is zero iff $\pi_\theta = \xi$. Any predictor that fulfills this is Bayes-optimal for $\xi$, and Eq. (1) and Eq. (2) provide a recipe for constructing an explicit Bayesian predictor, which is a mixture over one predictor per task, weighted by the posterior probability over the task given the context so far: $P(\tau|x_{<n}) \propto P(x_{<n}|\tau)P(\tau)$. In many cases this recipe is analytically or computationally intractable. Memory-based meta-learning provides an alternative for obtaining an approximate Bayesian predictor simply through log loss minimization in a meta-learning loop:

$$\arg\min_\theta D_{\mathrm{KL}}(\xi||\pi_\theta) = \arg\min_\theta \mathbb{E}_\xi\left[\log \frac{\xi(x_{1:N})}{\pi_\theta(x_{1:N})}\right] = \arg\min_\theta \mathbb{E}_\xi[\mathscr{L}_\theta(x_{1:N})] \tag{5}$$

If $\pi$ is expressive enough (realizability) and the meta-learning process fully converges, then, denoting $\hat\theta$ as the converged parameters:

$$\forall n \in \mathbb{N}^+ : \pi_{\hat\theta}(x_n|x_{<n}) \approx \xi(x_n|x_{<n}) \tag{6}$$

meaning the network's prediction over the next token, given context $x_{<n}$, is (nearly) indistinguishable from an explicit Bayesian predictor. The meta-trained neural network thus implements a Bayes-optimal adaptive prediction algorithm via its activations only—without weight updates (in-context learning). Previous works have empirically verified that meta-trained LSTMs and Transformers can indeed reach Bayes-optimality through meta-training, e.g., Mikulik et al. [2020] for sequential prediction and decision-making (not covered in this paper, but the theory extends straightforwardly to loss functions other than log loss), Genewein et al. [2023] for piecewise stationary data sources (where models additionally have to infer task boundaries), and Grau-Moya et al. [2024] for variable-order Markov processes.

**Remark.** Frontier models are *implicitly* meta-trained on samples from an unknown, rich and complex distribution of data generators [Chan et al., 2022a]. The main concerns w.r.t. the applicability of the Bayesian viewpoint is that due to limited expressivity, limited data, suboptimal optimization, and off-distribution inputs, models may not converge to or operate in the Bayesian regime. Additionally, models may often operate in the generalization regime, while the theoretical guarantees only hold strictly under data drawn from the training meta-distribution. While we strongly recommend carefully investigating these issues, the theory tells us that as models get better and better, they will get closer and closer to the Bayesian ideal, making it an important fundamental computational mechanism to understand.

# 3 Prompt Optimization: prefix-tuning

We are given a neural sequential predictor $\pi_\theta$, that was pretrained[2] via meta training over $\xi^{\text{Pre}}(x_{1:N}) = \int P(x_{1:N}|\tau)P^{\text{Pre}}(\tau)d\tau$. The goal is to adapt this predictor to a target distribution $\xi^{\text{Target}}(x_{1:N}) = \int P(x_{1:N}|\tau)P^{\text{Target}}(\tau)d\tau$. In prefix-tuning, the adaptation is performed by finding a (typically short) prefix sequence sequence $s_{1:L} \in \mathcal{S}^L$ of length $L$ that is prepended to the observations fed to the model. The "alphabet" $\mathcal{S}$ depends on the prefix-tuning method. In this paper we use:

- Hard token search (**HardPT**): $\mathcal{S} = \mathcal{A}$.
- Simplex prefix (**SimplexPT**): $\mathcal{S} = \Delta A \subset \mathbb{R}^{|A|}$.
- Real prefix (**RealPT**): $\mathcal{S} = \mathbb{R}^{|A|}$.
- Soft Prompting (**SoftPT**): $\mathcal{S} = \mathbb{R}^{\text{Embedding-dimensionality}}$, i.e., Lester et al. [2021].

Prefixing a sequence $x_{<n}$ with $s_{1:L}$ and passing it through the neural sequential predictor corresponds to conditioning with additional initial information: $\pi_\theta(x_n|s_{1:L}x_{<n}) = P_\theta(x_n|s_{1:L}, x_{<n})$. The prefix is optimized by minimizing the empirical log loss over $K$ samples of sequences from the target distribution *given* the prefix:

$$\min_{s_{1:L}\in\mathcal{S}^L} \mathbb{E}_{\xi^{\text{Target}}}\left[\mathscr{L}_\theta(x_{1:N}|s_{1:L})\right] \approx \min_{s_{1:L}\in\mathcal{S}^L} \frac{1}{K}\sum_{k=1}^{K}\left[\sum_{n=1}^{N} -\log P_\theta(x_n^k|x_{<n}^k, s_{1:L})\right] \quad (7)$$

with $x_{1:N}^k \sim \xi^{\text{Target}}$. For hard token search, we perform exhaustive search over all token sequences of length $L$. For all three soft token methods, we use mini-batch based stochastic gradient descent.

**When can prefix-tuning work?** We consider the case where the prefixed model behaves (near) Bayes-optimally on the target task distribution. We first consider the idealized Bayesian predictor for which the prefix is always a hard token sequence. A theoretical positive statement is possible if:

$$P^{\text{Target}}(\tau) = \delta(\tau = \tau^{\text{Target}}) \text{ and } P^{\text{Pre}}(\tau^{\text{Target}}) > 0 \quad (8)$$

that is, we are optimizing for a single target task that had support under the pretraining distribution. In this case, there always exists a sequence of hard tokens that causes the Bayesian posterior to concentrate sufficiently (in the limit a delta, see Appendix E ) for optimal prediction after the prefix. For sufficiently large $L$,

$$\exists s_{1:L} \in \mathcal{A}^L : \mathbb{E}_{P(x_{1:N}|\tau^{\text{Target}})}\left[-\log P(x_{1:N}|\tau^{\text{Target}})\right] \approx \mathbb{E}_{P(x_{1:N}|\tau^{\text{Target}})}\left[-\log \underbrace{\xi^{\text{Pre}}(x_{1:N}|s_{1:L})}_{\approx \pi_{\hat\theta}(x_{1:N}|s_{1:L})}\right] \quad (9)$$

The prefix can be found by performing the minimization in Eq. (7). Proof sketch: If the target distribution is a delta over one of the pretraining tasks (condition in Eq. (8)), then the argument that is being minimized in Eq. (7) is the r.h.s. of Eq. (9). The minimum is obtained when Eq. (9) becomes an equality, which the case when the Bayesian (posterior) mixture $\xi^{\text{Pre}}(\cdot|s_{1:L})$ collapses to a single mixture component corresponding to $\tau^{\text{Target}}$. This also implies $D_{\text{KL}}(\xi^{\text{Target}}||\xi^{\text{Pre}}(\cdot|s_{1:L})) \approx 0$ which is only possible iff $\xi^{\text{Target}} \approx \xi^{\text{Pre}}(\cdot|s_{1:L})$.

If the condition in Eq. (8) does not hold, optimal prompting may still be possible, but this strongly depends on the relationship between pretraining and target distribution and the model class. See Appendix E for an analysis of the Beta-Bernoulli case and beyond, including limits of prompting universal predictors. It is also possible to formulate two general theoretical negative cases:

**Prefix-tuning limitation I (multimodal target distributions).** In the limit posteriors can often not remain (or become) multimodal. If the target distribution is multimodal, such as a finite target mixture over tasks, optimal prefix-tuning may not be possible, even if all mixture components have support under the pretraining distribution. For instance, if the prior $P^{\text{Pre}}(\tau)$ is log-concave and the likelihood function is log-concave, then the posterior is also log-concave, and thus unimodal. See Section 4.2 for an empirical demonstration. More generally, if the posterior collapses to a Dirac delta in the limit (as it does for a Beta-Bernoulli model, and is very likely to do in general if prompts are typical sequences—see Appendix E), the only target task distributions that are optimally promptable are deltas over a single pretraining task.

---

[2]Except for our experiments with untrained networks, where the parameters are at random initialization.

**Prefix-tuning limitation II (novel atomic target tasks).** The second negative case is when the target distribution contains one or more novel atomic tasks, i.e., $P^{\text{Pre}}(\tau^{\text{Target}}) = 0$, that are "substantially" novel in the sense that they require behavior different from any of the predictors in the pretraining mixture $\xi^{\text{Pre}}$. E.g., a particular coin bias that was never observed during pretraining under a uniform bias over coins would not fall under this case. Note that it may be hard to say what counts as substantially novel at frontier model scale, where the pretraining distribution is only known implicitly, and pretrained models are capable of sophisticated algorithmic prediction. See Petrov et al. [2024] who, in line with our reasoning, find empirically that prefix-tuning methods can "elicit skills present in the pretrained model", but cannot be used to learn novel skills.

**Soft prefix-tuning.** In general, for a prefix prompt to be optimal, the model's internal state after consuming the prefix needs to be a sufficient statistic for the target distribution, without causing subsequent internal dynamics to diverge. Pretraining determines a model's state-update function, and thus imposes strong constraints w.r.t. possible manipulations via hard token inputs. These constraints can be partly overcome by using soft prefixes instead of hard tokens. As our results show, these carefully tuned off-distribution inputs that lead to off-distribution internal states, can be used to very effectively steer pretrained, and even untrained neural predictors. The limits of this mechanistic aspect, outside the conceptual Bayesian theory, are currently unclear—it could be that soft prefixes can very flexibly "reprogram" a pretrained model to arbitrary target distributions. Empirically, we find this not to be the case: while Soft Prompting does consistently improve prediction performance on the target task, it is still bound by the theoretical limitations pointed out above and in detail in Appendix E: optimally adapting a predictor pretrained over uniform random coins to a mixture of two coins is not possible. How large the potential gains from Soft Prompting or other soft prefix-tuning methods can be is an empirical question. Finally, note that weight-based fine-tuning methods are able to modify the pretrained state-update and -readout mechanisms, which allows for more flexibility w.r.t. adapting a network to target distributions (see Section 4.2 for empirical demonstrations).

## 4 Experiments on Coin-Flip Sequences

We conduct a series of experiments where a neural network is first meta-trained over a pretraining distribution (Section 2) of coin flip sequences of length $N_{\text{pre}} = 100$, and then prompt-tuned (Eq. (7)) or weight-tuned (mini-batch based log-loss minimization) to a target distribution of coin flip sequences of length $N_{\text{tune}} = 50$. After tuning, we evaluate tuned models on 2048 sequences of length $N_{\text{Eval}} = 200$ from the target distribution. Choosing $N_{\text{Eval}} > N_{\text{tune}}$ also allows to study how the solutions of different tuning methods generalize beyond the tuning length. Across experiments we use three different data distributions, two neural architectures, and nine tuning methods, which we now describe.

### 4.1 Experimental setup

**Data generators.** We use coin-flip sequences $P(x_{1:N}|\tau) = \text{Bernoulli}(\tau)$ with three different distributions $P(\tau)$ throughout our experiments. ***Random coins:*** $P(\tau) = \text{Beta}(1, 1)$, leading to a uniform distribution over coin biases. This is our pretraining distribution. The exact Bayesian predictor in this case is the Laplace predictor. ***Single coin:*** A single coin with bias $0.2$. This target distribution fulfills the condition that makes optimal prompting possible in Eq. (8). ***Two-coin mixture:*** A mixture of two coins, one with bias $0.2$ and one with bias $0.8$, with equal mixing weights of $1/2$ each. This target distribution violates the condition that theoretically allows for optimal prompting in Eq. (8). All tasks have binary outcomes which leads to a 2-dimensional one-hot token alphabet $\mathcal{A}$ with two different symbols. The Bayes predictors for all three tasks are analytically tractable and textbook examples (for the 'Single coin', the "Bayes" predictor is simply a constant probability).

**Neural sequential predictors.** We evaluate both LSTMs and Decoder-only Transformers. To support all fine-tuning methods we always use an initial embedding, and a final unembedding layer. The embedding is a trainable linear projection from the 2D token space into a 128-dimensional "embedding" space (results for 4-dimensional embeddings are shown in Appendix I). The unembedding is a trainable linear projection from the outputs of the final network layer down to the 2D logits. *Implementation Details.* The LSTM has a single hidden layer of width 128; the Transformer has a single multi-head attention layer with output dimensionality of 128, 4 attention heads, causal masking, SinCos positional encoding, a widening factor of 4 for the MLP block, and layer normalization after

query and key dense layers. Results for larger networks are shown in Appendix J. LoRA fine-tuning [Lester et al., 2021] is only supported for the Transformer, where we apply LoRA to all dense matrices of the attention block (query, key, value, and final attention weights). To produce a prediction given the empty context, we pass an initial zero vector $P_\theta(x_1|\epsilon) = \pi_\theta(x_1|\mathbf{0})$. This zero vector $\mathbf{0}$ is also prepended *before* any tunable prefix. When reporting the internal state of the LSTM, we use the cell state (hidden state gives qualitatively similar results), and for the Transformer we use the causally masked output of the attention block.

**Performance Measure.** Our main performance measure is the expected *cumulative regret*, which is the excess log loss compared to the ground-truth data generating probability. In Eq. (4) we defined the expected excess log loss relative to the best predictor that does not know $\tau$, a.k.a., the Bayesian regret. Similarly, we now define the excess log loss w.r.t. the data generator, that is, an oracle predictor that knows $\tau = \tau^*$:

$$\mathscr{R}_{\tilde{\theta}}^{P^{\text{Target}}}(N) := \mathbb{E}_{\tau^* \sim P^{\text{Target}}(\tau)}\mathbb{E}_{P(x_{1:N}|\tau^*)}\left[-\log \pi_{\tilde{\theta}}(x_{1:N}|s_{1:L}) + \log P(x_{1:N}|\tau^*)\right] \geq 0 \quad (10)$$

We show regret curves from $N = 0$ up to $N = N_{\text{eval}} - 1 = 199$ steps[3]. For prefix-tuning methods $\tilde{\theta}$ refers to the pretrained weights (or randomly initialized weights in our experiments on untrained networks) and $s_{1:L}$ is the tuned prefix. For weight-tuning methods $\tilde{\theta}$ refers to the tuned weights and the prefix is empty ($L = 0$). We (Monte-Carlo) estimate the regret with 2048 sequences sampled from the target data generator (from which we also get the ground-truth generating probabilities).

**Training and tuning details.** We pretrain for 1000 gradient steps (batch size 256, sequence length $N_{\text{pre}} = 100$, learning rate 0.001, and gradient clipping if the norm is $\geq 1$). For tuning we use 1000 steps (batch size of 256, thus $K = 256,000$, sequence length $N_{\text{tune}} = 50$, learning rate of $5e^{-3}$, and gradient clipping if the norm is $\geq 1$). We show tuning loss curves (and their convergence) in the extended results in the appendix. We repeat tuning runs 10 times per method with a different random seed (for sampling from the target distribution, and a different prefix initialization). Across repetitions and tuning methods, we always evaluate on the same set of 2048 evaluation sequences. Results are reported as the median over repetitions with $25\%, 75\%$ quantiles as "error bars".

**Fine-tuning methods.** We compare four different prefix-tuning methods and five different weight-tuning methods against a number of baselines:

- *HardPT, SimplexPT, RealPT, SoftPT:* prefix-tuning methods (see Section 3). To implement $\mathcal{S} = \Delta\mathcal{A}$ for SimplexPF, we pass the tunable prefix through a softmax. The prefix length $L = 6$ in all main experiments, and we show $L = 25$ in Appendix H.

- *EmbedWT, UnembedWT, Un+EmbedWT:* Only parameters of the linear embedding, or unembedding, or both, are tuned.

- *FullWT:* All weights, including embedding and unembedding, are tuned.

- *LoRAWT:* Low-Rank Adaptation [Hu et al., 2021], where an additive tunable low rank matrix (rank=4 in our experiments) is added to all dense matrices of the attention block. All other weights remain frozen.

- *TargetBayes, PreBayes, PreBayesPT:* exact Bayes predictors for the target distribution $\xi^{\text{Target}}$, the pretraining distribution $\xi^{\text{Pre}}$, and $\xi^{\text{Pre}}(\cdot|s_{1:L}^{\text{Target}})$, i.e., 'PreBayes' prefix-tuned to the target distribution via exhaustive hard token search, $L = 6$.

- *NoTuning:* the network with no fine-tuning. Either pretrained or at random initialization on our experiments with untrained networks.

- *RandomPF:* Same as 'NoTuning' but with random (one-hot) prefixes.

## 4.2 Results

**Tuning to a single task.** Fig. 1 shows that both, a Transformer and an LSTM, pretrained on Random Coins, can be Soft Prompted to be Bayes-optimal on a Single Coin (with bias 0.2). Other

---

[3]The offset of 1 is because passing $x_{1:n-1}$ through the network produces a prediction $x_n$ for which we need ground-truth data at step $n$ to compute the regret or log loss (or gradients).

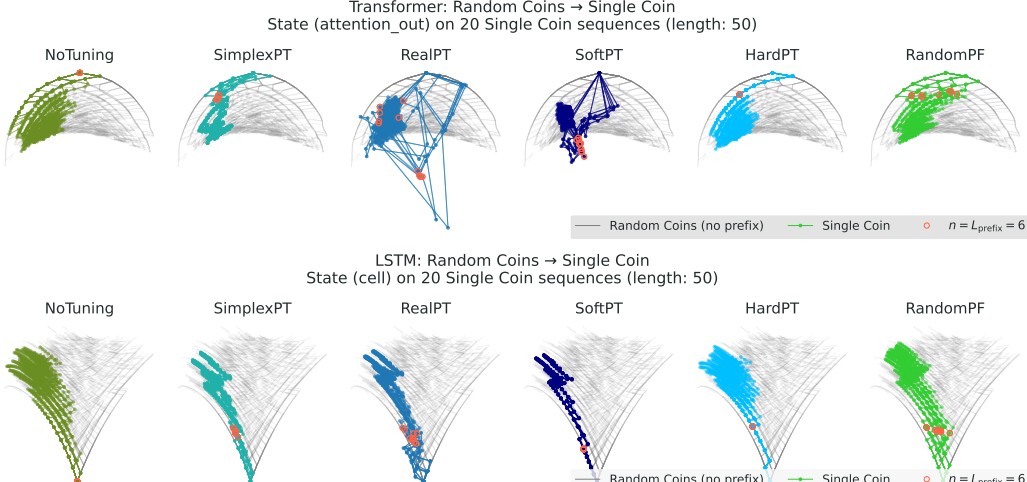

Figure 2: 2D PCA projection of Transformer's (top) and LSTM's (bottom) internal state (= activations), illustrating how differently tuned prefixes affect state and subsequent dynamics. Fig. A7 shows that the vertical principal component corresponds to the step $n$, and the horizontal to the heads-to-tails ratio. Colored lines are sequences from the target distribution (single coin with bias $0.2$), gray lines are from the pretraining distribution (uniform random). The off-distribution nature of soft prefixes is particularly visible for the Real- and Soft-prefix for the Transformer. See Fig. 1 for regret curves.

prefix-tuning methods, including exhaustive search over all hard token sequences of length 6 fail to reach Bayes-optimality. Despite all soft prefixes being off-distribution inputs, internal dynamics remain stable (see Fig. 2), and prediction generalize well for 'SoftPF' and most of the weight-tuning methods far beyond the tuning sequence length $N_{\text{tune}} = 50$ (see Fig. 1). The results illustrate how prefixes can be used to steer a (meta-learned) Bayesian predictor via manipulating its internal state. Further, they also show that off-distribution inputs can be particularly effective; more effective than even the best possible hard token sequence of the same length.

Note that for all prefix-tuning methods the dimensionality of $\mathcal{S}$ is the same as the one-hot token alphabet (i.e., 2-dimensional for coin-flip tasks), except for 'SoftPT' where the prefix embeddings of dimensionality $128$ are tuned. These additional degrees of freedom are the main source of superior performance in our experiments, and we demonstrate that the advantage largely disappears when reducing the embedding dimensionality to $4$ in Appendix I. Since LLMs typically have a larger input-than embedding dimensionality, tuning inputs ('RealPT') may be as efficient, or even more efficient, compared to tuning embeddings ('SoftPT') for LLMs.

**Limitation: prompting to task mixtures.** Fig. 3 empirically demonstrates the theoretical shortcoming of prefix-tuning discussed in Section 3: prompt tuning to a mixture of two coins is not possible if the pretraining distribution is uniform random coins. While Soft Prompting, being the strongest prefix-tuning method, leads to performance gains compared to the untuned pretrained predictor, it is not enough to reach 'TargetBayes' on the Two-Coin Mixture. As expected, some weight-tuning methods can lead to that level of performance, at the cost of permanently altering the pretrained predictor. While soft prefixes are strictly speaking not covered by the Bayesian theory (because they are off-distribution inputs exploiting the circuitry of the particular pretrained network), these results highlight the importance of the Bayesian view in practice: in the absence of theoretical understanding it might have been quite puzzling why one can optimally prefix prompt for a single coin but not a mixture of two coins. We confirm that the result is not an artifact of limited soft prefix length—see Fig. A10 for a control experiment with $L = 25$, and Fig. A12 for results with larger networks.

**Comparing prefix- and weight-tuning for untrained networks.** Fig. 4 shows that it is possible to Soft Prompt an untrained Transformer quite well to the Two-Coin Mixture and Random Coins as target tasks, meaning that relatively complex in-context algorithms are easily available in the untrained net. There is still a gap to 'TargetBayes' performance though. Results in the Appendix (Figs. A8 and A9) show that Soft Prompting the untrained LSTM has very little effect, indicating

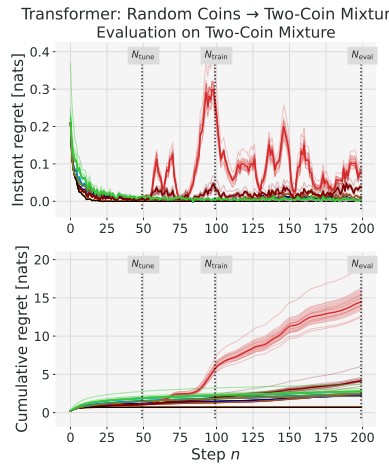

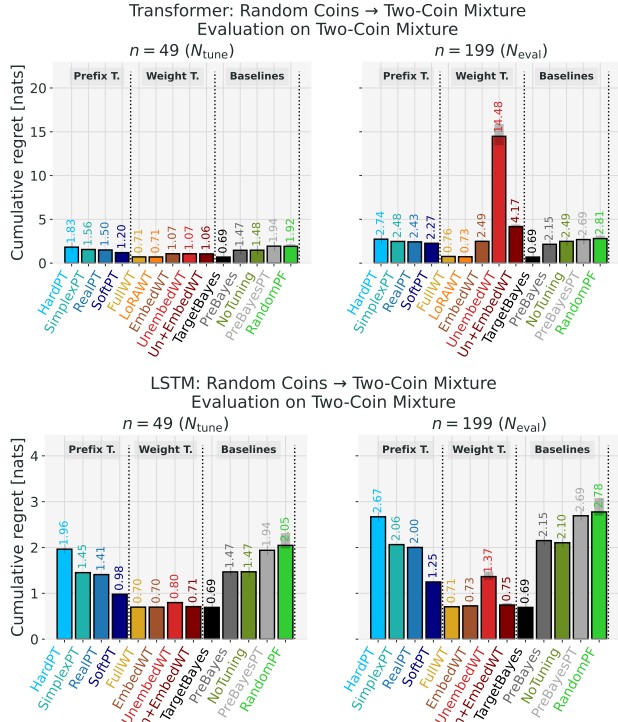

**Top:** Performance of different tuning methods, measured as excess log loss, i.e., regret (Eq. (10), lower is better). See bar plots for color legend. **Top-right:** Detailed Transformer results for last step within the tuning sequence length $N_{\text{tune}}$ and the last evaluation step $N_{\text{eval}}$. **Right:** Like above but for LSTM.

Figure 3: Models pretrained on sequences from coins with uniform random bias (length $N_{\text{train}} = 100$) are fine-tuned to the target task of a mixture of two coins (tuning sequence length $N_{\text{tune}} = 50$). No prefix-tuning method (with prefixes of length 6) can achieve optimal performance on the target task ('TargetBayes' is optimal). Full weight-tuning, LoRA (on the Transformer) and two of the embedding tuning variants on the LSTM do reach optimality (even beyond the tuning length of 50 steps). See Fig. A7 for a visualization of how different prefixes affect models' internal dynamics. Regret curves for the LSTM, similar to Top left panel, are shown in Fig. A6.

a fundamental difference between the Transformer and the LSTM in this regard (which behave very similarly on our experiments when pretrained). See also Zhong and Andreas [2024], who tune untrained Transformers to algorithmic tasks via embedding- or unembedding-tuning, or both. They find that, tuning both the embedding and unembedding is important on their tasks. If our results qualitatively hold at their tasks, then Soft Prompting of untrained networks should be only slightly worse, and LoRA should perform even better than Un+Embedding tuning. As previously, note that the superiority of 'SoftPT' in our setting is largely explained by having a much higher embedding dimensionality compared to the input dimensionality (which is typically reverse at LLM scale). Increasing the soft prefix length to $L = 25$ does not significantly improve performance, see Fig. A10.

## 5 Discussion

While we have laid important fundamental groundwork in our current study, extrapolating our findings to modern frontier model (and data) scale is not straightforward. The theoretical findings we presented, including the limitations of prefix tuning, hold at any scale, but it is likely that additional practical issues arise at large scale that are not captured by our current experiments. It is thus hard to predict the relevance and impact of our fundamental results on today's frontier-model practice. For instance, one of our main results is that optimal prompting to a single target task is possible, whereas it is not for a mixture of tasks. We are confident that this holds even at frontier model scale (based on the theory), but it is unclear and highly non-trivial what constitutes a task for a LLM, and accordingly, whether this is a severe limitation or not. In our experiments, a task is simply an unobserved variable in a two-level hierarchical statistical model. At LLM scale, the structure is vastly more complex, with many more hierarchical levels, and potentially other statistical structures at play.

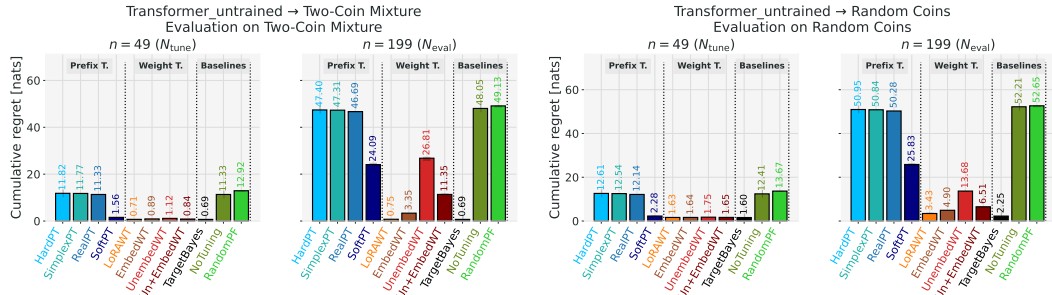

(a) Untrained Transformer tuned to Two-Coin Mixture.    (b) Untrained Transformer tuned to Random Coins.

Figure 4: Untrained Transformer tuned to the Two-Coin mixture (left) and to Random Coins (right); tuning sequence length $N_{\text{tune}} = 50$. In both cases, Soft Prompting is the only effective prefix-tuning method. It nearly reaches Bayes-optimality ('TargetBayes', which is a Laplace predictor on Random Coins). Performance degrades rapidly after the tuning sequence length. Full regret curves (and LSTM results) in Fig. A8 and Fig. A9. Among the weight-tuning methods, LoRA is very effective.

The next step would be to carefully design data generators that are closer to natural language data, but still fully understood and well controllable, akin to the data generators used in [Allen-Zhu, 2024], and run our experiments of tuning to single tasks vs. tuning to mixtures of tasks at scale. With these caveats in mind, cautiously extrapolating our findings to frontier model scale, raises some questions for investigation, which we now discuss.

Given our results, soft prefix tuning should be superior to tuning sequences of hard tokens—if a sufficiently large fine-tuning set is available, soft prompt tuning should beat prompt engineering, and other methods of hard token optimization such as PromptBreeder [Fernando et al., 2023][4]. Similarly, instead of conditioning on a large set of in-context examples for imitation learning (e.g., Ruoss et al. [2025], Paglieri et al. [2025]), it may be beneficial to distill these examples into a more effective tuned soft prefix. The superiority of Soft Prompting over other soft prefix-tuning methods is largely due to the much higher dimensionality of "embeddings" compared to the input space in our experiments (see control experiments with embedding dimensionality 4 in Appendix I). This is typically reversed in frontier models, which could mean that soft input tuning is more effective than embedding tuning.

Finally, our experiments raise the question: why prompt (-tune) at all, when weight-based methods, such as LoRA, are equally or more effective and do not suffer from the theoretical limitations pointed out? First, weight-tuning permanently alters a network and would lead to performance decreases on the pretraining distribution (which can be overcome by storing the set of original weights). Additionally, comparisons between in-context and in-weight learning at LLM scale find that in-weight learning can sometimes be very limited and generalize poorly [Lampinen et al., 2025, Chan et al., 2022b]. Since prompting fundamentally builds on a network's in-context adaptation mechanisms, it may be the case that prompt-tuning works better than weight-tuning in cases where in-context learning generalizes better than in-weight learning. An interesting future research question is whether tuned (soft) prefixes transfer between different models (perhaps with additional regularization)—if true, the prefix-tuning cost would only have to be spent once, compared to weight-tuning methods that need to be run for every network to fine-tune. Please see our discussion of additional related work in Appendix D, where we discuss a number of previous works that investigate in-context learning under a Bayesian and/or meta-learning lens, such as Xie et al. [2021], Kirsch et al. [2022], Lampinen et al. [2024], and Elmoznino et al. [2024].

To conclude, the Bayesian view on prompt tuning, which arises from analyzing memory-based meta-learning, provides a conceptual understanding that leads to a formal characterization of prompting and some of its fundamental limitations. We have shown that these limitations hold in practice, for both hard token prefix-tuning, but also when optimizing soft prefixes—a setting not fully covered by the theory (which does not consider non-token inputs). The code to reproduce all our experiments and figures is available at: `https://github.com/google-deepmind/thunnini`.

---

[4]Though some hard-token tuning methods may have the advantage of resulting in more interpretable prompts.

## Acknowledgments and Disclosure of Funding

We thank Joel Veness, Gregoire Deletang, Satinder Baveja and Shane Legg for helpful feedback and discussions.

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

## A    Societal impact

While LLMs and frontier models have broad and significant societal impacts today, and increasingly so in the future, our work aims at understanding one of the fundamental conceptual mechanisms w.r.t. how rapid in-context learners can be steered via prompt optimization. Our analysis is theoretical, our experiments are educational, and we do not propose novel, more powerful methods. This fundamental understanding may enable the development of better methods to steer frontier models more precisely and more data efficiently, which could further boost societal impacts (both positive and negative) and facilitate abusing or attacking models via prompts as well as defending and hardening against such attacks with better system prompts. Weighing up these (hypothetical) factors, we firmly believe that better understanding generally leads to more robust and safe technology.

## B    LLM usage

There was no LLM use involved in authoring this paper and its experiments. No part of this paper and the accompanying code was authored or modified by, or inspired through conversations with an LLM. Smart auto-complete was used when writing code, which is partly powered by a coding model, but no LLM or coding model was explicitly prompted to (co-)author or edit parts of the codebase.

## C    Compute usage

The educational experiments presented in the paper were run on a single V100 GPU in under 6 hours.

## D    Additional Related Work

In-context learning has been studied extensively in the recent literature. In many cases, it specifically refers to a particular type of supervised few-shot learning. In contrast, Lampinen et al. [2024] (among others) argue that a whole number of LLM in-context abilities can be unified as in-context learning in a wider sense: "[...] *we suggest that any distribution of sequences in which context non-trivially decreases loss on subsequent predictions can be interpreted as eliciting a kind of in-context learning. We suggest that this perspective helps to unify the broad set of in-context abilities that language models exhibit.*". This is in line with our view on in-context learning, and we argue that the Bayesian perspective that arises from analyzing memory-based meta-learning provides the unifying theoretical framework. Lampinen et al. [2024] also discuss memory-based meta-learning as the underlying factor. We present this connection in more formal detail (see Section 2), and use it to drive the design of our experiments. Bayesian inference has also been put forward as an explanation for the mechanism that drives in-context learning in Xie et al. [2021], Müller et al. [2022], Genewein et al. [2023], Binz et al. [2024], Wang et al. [2023], Panwar et al. [2024].

The theoretical aspect that ties together meta-learning, Bayesian inference, and optimal prompting is minimization of prediction error (log loss). A dual, and fully equivalent view is maximizing a (lossless) compression objective. Deletang et al. [2024] discuss this well-known duality [MacKay, 2003] in the context of language modeling and show that pretrained LLMs are surprisingly good compressors for image and audio data. This is further expanded by Heurtel-Depeiges et al. [2024], who show that (medium-sized) pretrained transformers' in-context learning abilities can lead to lossless compression on par with general-purpose compression algorithms, such as gZip, across different modalities. For a great recent theoretical discussion on in-context learning and how it arises from meta-training and relates to algorithmic compression, see Elmoznino et al. [2024]. Their theoretical discussion is complementary to ours: shifting to an algorithmic statistical view [Li et al., 2008, Hutter et al., 2024], as they do, allows to more easily make statements about generalization—in contrast, the classical statistical view requires distributions, which makes it harder to formally characterize off-distribution generalization. We have focused on the latter for simplicity and conciseness, but note that Bayesian inference and in-context learning carry over into algorithmic statistics by considering distributions over programs [Rathmanner and Hutter, 2011, Hutter et al., 2024]. The algorithmic, and classical Bayesian view are thus largely equivalent, and a simplicity prior similar to what is discussed in Elmoznino et al. [2024] also appears as an "automatic" Bayesian Occam's Razor [MacKay, 2003] in classical Bayesian inference and non-algorithmic minimum description-length (MDL)—though in the classical case the simplicity prior is not Kolmogorov complexity. The main point is that a

Bayesian mixture predictor over a large class of programs is not at odds with our investigation in this paper. For frontier model scale, such an intuition may be more appropriate, as it would imply that very complex *algorithms* can be executed in-context, including sophisticated learning algorithms (see also Schuurmans [2023], Schuurmans et al. [2024] for a discussion of how LLMs are universal in principle). This also means that more specific in-context learning algorithms can be identified in particular settings, without violating the Bayesian view—e.g., explanations of in-context learning as gradient descent [Von Oswald et al., 2023, Mahankali et al., 2024], linear and ridge-regression [Akyürek et al., 2023], and learning linear functions [Garg et al., 2022].

We explicitly meta-train on simple tasks for our experiments, which allow for the comparison against a known tractable Bayesian predictor, similar to Mikulik et al. [2020], Genewein et al. [2023], Wenliang et al. [2025]. Other works have also used simple synthetic examples to study in-context learning and develop algorithmic understanding [Akyürek et al., 2023, Garg et al., 2022, Mahankali et al., 2024, Elmoznino et al., 2024]. Beyond simple examples, explicit meta learning at scale can give rise to complex adaptive in-context algorithms. For instance, Bauer et al. [2023] meta-train an agent that adapts in-context at human timescales, in terms of number of interaction episodes, to a vast number of tasks in a simulated 3D environment. Another notable example is Laskin et al. [2023], who meta-train an in-context reinforcement learning algorithm (see also Wang et al. [2016]). While LLMs are not explicitly meta-trained, Chan et al. [2022a] argue that naturalistic data like language has many of the properties of meta-learning datasets and show that these properties drive in-context learning.

To adapt (large) pretrained models, many fine-tuning methods, with a number of variations each, have been proposed in the recent literature. See Han et al. [2024] for a review of parameter-efficient fine-tuning methods, such as, soft prompting [Lester et al., 2021], prefix prompting [Li and Liang, 2021], or LoRA, Hu et al. [2021]. Whether a method falls under prefix-tuning or weight-tuning may not always be immediately obvious, since tunable inputs also appear as indirectly tunable parameters in query and key matrices of transformers. Careful analysis reveals though, that tunable parameters resulting from prefix-tuning methods are more constrained compared to weight based tuning. Petrov et al. [2024] perform such an analysis and find: "[...] *while techniques like prompting, in-context learning, soft prompting, and prefix-tuning can effectively elicit skills present in the pretrained model, they may not be able to learn novel tasks that require new attention patterns.*". This empirical finding is in line with one of the theoretical negative results for prompting that we point out in Section 3. The difference between in-context learning (which is the mechanism that prompt tuning exploits) and in-weight learning has been studied in a series of works [Lampinen et al., 2025, Agarwal et al., 2024, Chan et al., 2022b] that find that in-context learning can typically generalize more flexibly than weight-tuning, and that the underlying mechanisms are quite different (e.g., rule-based vs. exemplar-based generalization), and may result from different neural circuits that compete during training [Singh et al., 2023, 2025], as well as different underlying properties of the data distribution [Chan et al., 2025]. Kirsch et al. [2022] study general-purpose in-context learning via meta-learning and investigate the factors that lead models to generalize (via meta-learned in-context algorithms) as opposed to memorization. An interesting approach presented in Bornschein et al. [2023] is to switch from in-context to in-weight learning as soon as the available examples allow it (in terms of number of samples and statistical properties w.r.t. the pretrained predictor), which is determined via prequential evaluation.

One large downside of soft prompts may be that hard token sequences are potentially more interpretable. The literature on interpretability and understanding of prompts and prompting techniques is growing rapidly and surfaces complex problems Patel et al. [2025], Bailey et al. [2023], Petrov et al. [2024].Particularly Bailey et al. [2023] find that soft prompts are generally hard to intrepret, also when trying to map them to hard token sequences. Su et al. [2022], Qin et al. [2021], Zheng et al. [2024] focus on understanding prompts through task sub-spaces and how they enable transfer between tasks. At least some of the issues may not be specific to soft prompts, as analyzed by Wenliang et al. [2025], which discusses a number of theoretical and fundamental practical issues with identifying and interpreting optimal hard-token prefixes, such as a high sensitivity of optimized prompts on the pretraining and target distribution (including aspects like sequence length and fine-tune set size). They also find some indication that LSTMs can have more easily interpretable hard prompts compared to Transformers.

# E  Theoretical limits of prompting Bayesian predictors

Even for idealized Bayesian predictors, and hard prefix prompting (i.e., no inputs outside the token alphabet), the theoretical limits of what can be achieved via prefix prompting are non-trivial. We first discuss the Bernoulli case, as it is most relevant to our experiments. We then extend the model class to show two constructions for which optimal prompting is possible—arguably the constructions are somewhat artificial, and do not relate to current paradigms of pretraining frontier models. We then ask whether optimal prompting to arbitrary target distributions is possible for a universal Bayesian predictor, i.e., the Solomonoff mixture [Rathmanner and Hutter, 2011, Grau-Moya et al., 2024]. The answer is yes, but optimal prompts may need to be very long (and grow in length with increasing approximation quality). Whether the Solomonoff mixture can be 'efficiently' prompted remains an open problem. Finally we show that on average prompting narrows the task distribution, but atypical prompts can widen it.

**Bernoulli mixtures.**  Let $\mu_\theta(x_{1:n}) := \theta^k(1-\theta)^{n-k}$ with $k_n := x_1 + ... + x_n$ and $x_t \in \mathbb{B} := \{0, 1\}$ be a Bernoulli($\theta$) process with prior (density) $w(\theta)$ for $\theta \in [0; 1]$. Then the posterior is $w(\theta|x_{1:n}) = \mu_\theta(x_{1:n})w(\theta)/\zeta_w(x_{1:n})$, where normalizer = mixture = evidence $\zeta_w(x_{1:n}) := \int_0^1 \mu_\theta(x_{1:n})w(\theta)d\theta$.

If $w(\theta)$ is a Beta prior (e.g. uniform), then the posterior is also Beta-distributed with variance $\mathrm{Var}[w(\cdot|x_{1:n})] \leq k(n-k)/n^3 \leq 1/4n \to 0$ for $n \to \infty$. Hence the posterior of $w$ "converges" to a $\delta$-peak whatever $x_{1:\infty}$.

More generally, for any prior such that $w$ and sequence $x_{1:\infty}$ for which all limit points of $\hat{\theta}_n := k_n/n$ are in the support of $w$, we have $\mathrm{Var}[w(\cdot|x_{1:n})] \to 0$. Note this holds even if $k_n/n$ itself does not converge.

Also, if the prior $w(\theta)$ is log-concave (e.g. uniform), the posterior $w(\theta|x)$ is also log-concave, and hence unimodal. In particular, in these cases there is no prefix $y$ such that the predictive distribution $\zeta_w(\cdot|y)$ is a mixture of two (or more) Bernoullis.

Situations for which the posterior does not collapse are rare and somewhat artificial: for instance if the prior $w(\theta) = \frac{1}{2}\delta(\theta - \frac{1}{3}) + \frac{1}{2}\delta(\theta - \frac{2}{3})$, i.e. $\zeta$ is a mixture of just two Bernoullis, and $k = n/2$, then the posterior is also a mixture over two Bernouellis $w(\theta|x_{1:n}) = w(\theta)$. Or if the prior has a gap: $w(\theta) = \frac{3}{2}[\![\theta \leq \frac{1}{3} \vee \theta \geq \frac{2}{3}]\!]$ and $k = n/2$, then the posterior will retain the gap and remain bimodal.

In summary: under many priors, including Beta priors, the posterior collapses to a delta (zero variance) under increasing observations. No prefix can thus lead to a posterior over a mixture of, say, two coins with different bias. Even if the posterior has not fully collapsed, starting with a log-concave prior can only lead to a unimodal posterior. Exceptions are technically possible, e.g., when the prior is already a mixture over two components or has a gap.

**Countable mixtures.**  One can ask whether a larger (countable) class of distributions $\mathcal{M} = \{\nu\}$ always allows for optimal prefix-tuning: Let the mixture $\xi(x) := \sum_{\nu \in \mathcal{M}} \nu(x)\tilde{w}(\nu)$ with some prior $\tilde{w}(\nu) > 0$. Indeed, for suitable $\mathcal{M}$ it holds that: for every computable (Bernoulli) prior $w$ there exists some $y \in \mathbb{B}^*$ such that $\xi(x|y) = \zeta_w(x)$ for all $x \in \mathbb{B}^*$. That is, one can prompt $\xi$ such that its predictive distribution becomes any desired Bernoulli mixture. The construction is quite artificial though: we require that the model class is such that the prefix is interpreted as a program that explicitly computes the desired target prior. Formally, let $\nu_p(x_{1:n}) := [\![x_{<\ell} = p]\!]\zeta_w(x_{\ell:n})$, where $p \in \mathbb{B}^*$ is a prefix program computing prior $w()$ and $\ell - 1 = $length$(p)$ (i.e. $\mathcal{P} := \{p : p$ computes some $w\}$ is a prefix-free set). Let $\mathcal{M} = \{\nu_p : p \in \mathcal{P}\}$. Then it is easy to see that $\xi(x_{\ell:n}|x_{<\ell}) = \zeta_w(x_{\ell:n})$, where $x_{<\ell} \in \mathcal{P}$ computes prior $w$.

This class is very artificial, but it shows that prefix-tuning for general Bernoulli mixtures is possible in principle. No special property of $\zeta_w$ was used, so the above tuning construction works for arbitrary countable base class $\mathcal{B} = \{\zeta_i\}$.

**Product mixtures.**  To overcome the need that the prefix is a program that computes the prior, another construction is possible, where the prompt is a number of increasingly longer sequences of samples from the target distribution. Let $\mathcal{B} = \{\zeta\}$ be a countable class of target distributions (e.g. Bernoulli mixtures). Let $1 = i_0 < i_1 < i_2 < ...$ be a sequence of temporal boundaries with increasing segment lengths $\delta_\kappa := i_\kappa - i_{\kappa-1} \to \infty$ for $\kappa \to \infty$, e.g. $\delta_\kappa = \kappa$ or $\delta_\kappa = 2^\kappa$. De-

fine the product distribution as $\nu_\zeta(x_{1:n}) := \prod_{\kappa=1}^m \zeta(x_{i_{\kappa-1}:i_\kappa-1})$ for $n = i_m - 1$, and for other $n$ by marginalization. Let $\xi(x) := \sum_{\zeta \in \mathcal{B}} \nu_\zeta(x)\tilde{w}(\zeta)$ be a Bayes mixture with some prior $\tilde{w} > 0$. Then by [Hutter, 2005, Sec.3.7.1] or Blackwell and Dubins [1962] for any $h < \infty$ we have $\xi(x_{\ell:\ell+h-1}|x_{<\ell}) \to \nu_\zeta(x_{\ell:\ell+h-1}|x_{<\ell})$ a.s. if $x_{1:\infty} \sim \nu_\zeta$. If $i_m - h \geq \ell = i_{m-1}$ for some $m$, then $\nu_\zeta(x_{\ell:\ell+h-1}|x_{<\ell}) = \zeta(x_{\ell:\ell+h-1})$. The condition is satisfied for infinitely many $\ell$. Hence for any $\zeta \in \mathcal{B}$ and any $h \in \mathbb{N}$ there exists a prompt $x_{<\ell}$ such that $\xi(x_{\ell:\ell+h-1}|x_{<\ell}) \approx \zeta(x_{\ell:\ell+h-1})$ for all $x_{\ell:\ell+h-1} \in \mathbb{B}^h$, and the approximation error can be made arbitrarily small by suitably large $\ell$.

Compared to the countable mixture construction from before the constructed class and prompt sampled from a product of the target distribution $\zeta$ are more natural. As we will see below, Solomonoff's $M$ can also be prompted in this way. The downsides of this construction are that the approximation $\xi(x|y) \approx \zeta(x)$ is non-uniform in the length of $x$ and longer $x$ require longer prompts $y$. Additionally, the required prompt $y$ is typically much larger than program prompt $p$ in the countable mixture construction above.

**Solomonoff mixture.** Finally, we ask whether optimal prompting to any target distribution is always possible for a universal predictor. Let $M$ be Solomonoff's a priori distribution, and $\zeta$ be some computable distribution, and $i_k$ be a computable index sequence. Then $\nu_\zeta$ is also computable, hence included in the mixture $M$ ($M(\cdot) \geq c \cdot \nu_\zeta(\cdot)$ for some constant $c > 0$). The same argument as before implies that $M(x|y) \approx \zeta(x)$ for suitable $y$. That is, Solomonoff's $M$ can be prompted to approximate any other computable distribution. However, this argument suffers from the same downsides regarding the length of the required prompt.

It is an open problem whether $M$ can be efficiently prompted similarly to the 'countable mixture' case with a short prompt $p$ and approximation accuracy uniform in the length of $x$.

**Entropic analysis.** In expectation, extra information decreases entropy ($H(X|Y) \leq H(X)$), but specific information can increase or decrease entropy ($H(X|Y = y) \gtrless H(X)$). In our sequential context this means that $H(\cdot|X_{<\ell}) \leq H(\cdot)$, where $\cdot$ can be $X_{\ell:\infty}$ or $X_{\ell:\ell+h-1}$ or $\theta$. This means under some ergodicity assumptions for large $\ell$, if $x_{<\ell} \sim P = \xi$, then likely $H(\cdot|x_{<\ell}) \lesssim H(\cdot)$, i.e. typical prompts narrow the task distribution. In the Beta-Bernoulli case we even have $H(\theta|x_{<\ell}) \to -\infty$ if $x_{1:\infty} \sim \xi$ whatever the prior $w(\theta)$, and the posterior converges to a $\delta$-peak. Conversely if we want the posterior $w(\theta|x_{<\ell})$ to be broader (have higher entropy) than the prior $w(\theta)$, we need an atypical prompt $x_{<\ell}$. For instance, for a Beta-Bernoulli with prior $w(\theta) = \varepsilon\delta(\theta) + (1-\varepsilon)\delta(\theta - \frac{1}{2})$ and small $\varepsilon$, the entropy $H(\theta|x_{<\ell})$ increases with $\ell$ for small $\ell$ for the atypical prompt $x_{<\ell} = 0^{\ell-1}$.

# F    List of Notation

| Symbol | Explanation |
|---|---|
| $\mathcal{A}$ | Token alphabet (in our case binary one-hot tokens, i.e., $|\mathcal{A}| = 2$) |
| $\Delta\mathcal{A}$ | Probability vector over a token from the alphabet. |
| $x_{1:N} \in \mathcal{A}^N$ | Sequence of tokens from alphabet $\mathcal{A}$ of length $N$. |
| $N_{\text{train}}$ | Pretraining sequence length $= 100$. |
| $N_{\text{tune}}$ | Tuning sequence length $= 50$. |
| $N_{\text{eval}}$ | Evaluation sequence length $= 200$. |
| $\epsilon$ | The empty sequence. |
| $s_{1:L} \in \mathcal{S}^L$ | Prefix of length $L$. |
| $L$ | Prefix length $= 6$ (or 25 for control experiments). |
| $\mathcal{S}$ | "Alphabet" for prefix. Depends on tuning method. |
| $\tau \in \mathbb{R}^M$ | $M$-dimensional parameter vector of a task. |
| $P(\tau)$ | Task distribution. |
| $P^{\text{Pre}}(\tau)$ | Pretraining task distribution. |
| $P^{\text{Target}}(\tau)$ | Pretraining task distribution. |
| $P(x_{1:N}|\tau)$ | Distribution over sequences induced by task. Function from $\mathbb{R}^M \to \Delta\mathcal{A}^N$. |
| $\xi(x_{1:N})$ | Marginal distribution over sequences $= \int P(x_{1:N}|\tau)P(\tau)d\tau$, |
| | also: Bayes mixture with prior $P(\tau)$, |
| | also: Bayes predictor for task distribution. |
| $\xi^{\text{Pre}}$ | Pretraining sequence distribution / Bayes predictor. |
| $\xi^{\text{Target}}$ | Target sequence distribution / Bayes predictor. |
| $\pi_\theta$ | (Neural) parametric sequential predictor. Function from : $\{\mathbb{R}^D\}^* \to \Delta\mathcal{A}$. |
| $P_\theta$ | Distribution over tokens induced by (forward passes through) $\pi_\theta$. |
| $D$ | Dimensionality of inputs for neural net. $D = |\mathcal{A}| = 2$ in our case. |
| $\theta$ | Parameters of neural sequential predictor. |
| $\hat{\theta}$ | Parameters converged to optimum after pretraining. |
| $\tilde{\theta}$ | Parameters after prefix- or weight-tuning to compute regret |
| | $= \hat{\theta}$ if prefix-tuning and net is pretrained optimally, |
| | $=$ at random initialization if prefix-tuning and net is untrained, |
| | $=$ tuned weights if weight-tuning. |
| $\xi^{\text{Pre}}(\cdot|s_{1:L}^{\text{Target}})$ | Bayes predictor for $\xi^{\text{Pre}}$, prefix tuned to Target distribution. |
| $\mathcal{L}_\theta(x_{1:N})$ | Log-loss for single sequence. |
| $\mathcal{L}_\theta(x_{1:N}|s_{1:L})$ | Log-loss for single sequence prefixed by $s_{1:L}$ (loss only over $x_{1:N}$). |
| $\mathcal{R}_{\tilde{\theta}}^{P^{\text{Target}}(\tau)}$ | Cumulative regret of net with parameters $\tilde{\theta}$ on target distribution. |
| $K$ | Number of sampled sequences for log loss minimization during tuning. |

# G    Full results

Some of the following results are included in the main paper, for completeness we now show all results per experiment and network architecture. For all regret curves and bars, thick lines or bars show the median over 10 fine-tuning repetitions, thin lines show individual repetitions, and shaded areas or bars show $25\%, 75\%$ quantiles as confidence intervals.

### G.1 Petraining on Random Coins, fine-tune to Single Coin

Full details on the regret curves and tuning loss curves are given in Fig. A5, and the impact of tuned prefixes on the networks' internal dynamics is shown in Fig. 2. To interpret the latter, see Fig. A7, which shows that the internal state of models pretrained on Random Coins is highly structured, with one of the two principal components corresponding to the step $n$ and the other to the heads-to-tails ratio.

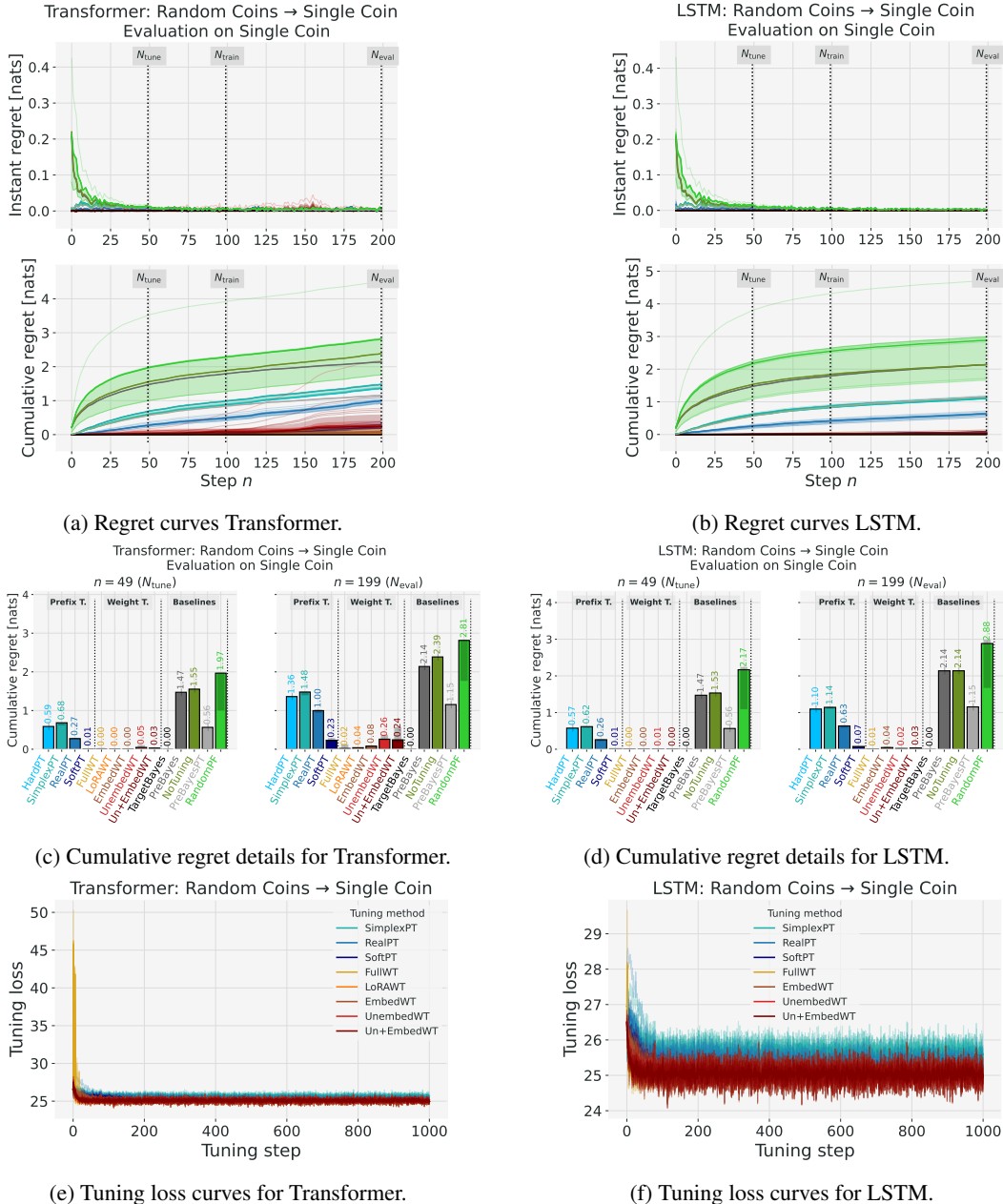

(a) Regret curves Transformer.

(b) Regret curves LSTM.

(c) Cumulative regret details for Transformer.

(d) Cumulative regret details for LSTM.

(e) Tuning loss curves for Transformer.

(f) Tuning loss curves for LSTM.

Figure A5: Models pretrained on Random Coins are tuned to a Single Coin. Transformer shown in the left column, LSTM shown in the right column. Of the prefix-tuning methods, only Soft Prompting ('SoftPT') allows optimal target task performance. Several of the weight-tuning methods succeed.

## G.2  Pretraining on Random Coins, fine-tune to Two-Coin Mixture

Full details on the regret curves and tuning loss curves are given in Fig. A6, and the impact of tuned prefixes on the networks' internal dynamics is shown in Fig. A7.

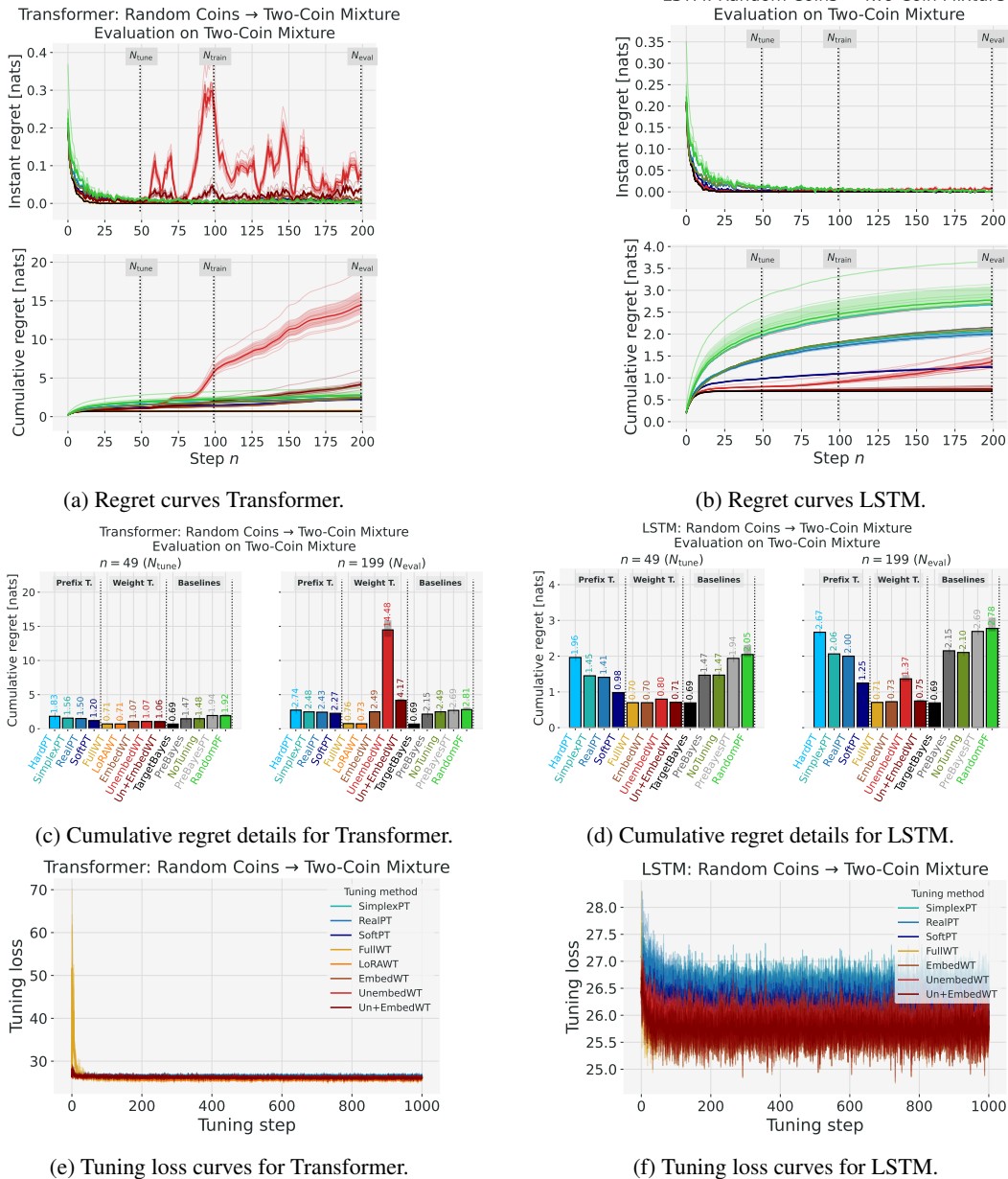

(a) Regret curves Transformer.

(b) Regret curves LSTM.

(c) Cumulative regret details for Transformer.

(d) Cumulative regret details for LSTM.

(e) Tuning loss curves for Transformer.

(f) Tuning loss curves for LSTM.

Figure A6: Models pretrained on Random Coins are tuned to a Two-Coin Mixture. Transformer shown in the left column, LSTM shown in the right column. No prefix-tuning method allows optimal target task performance (shown as 'TargetBayes'). Some of the weight-tuning methods succeed.

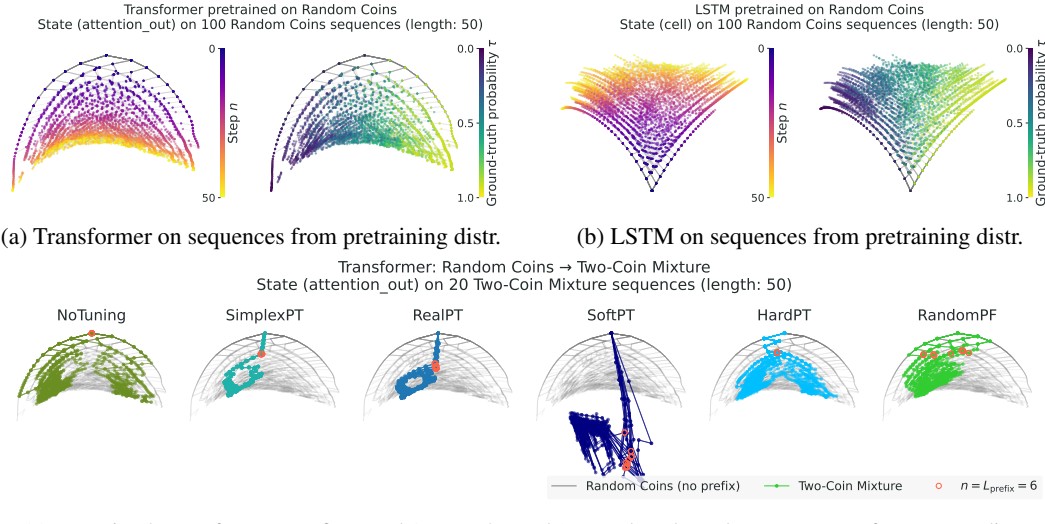

(a) Transformer on sequences from pretraining distr.

(b) LSTM on sequences from pretraining distr.

(c) Pretrained Transformer, prefix tuned (center 4 panels) to and evaluated on sequences from target distr.

(d) Pretrained LSTM, prefix tuned (center 4 panels) to and evaluated on sequences from target distr.

Figure A7: Top: Internal state (2D PCA projection) of pretrained models is highly structured—one component tracks the step $n$ and the other component tracks heads-to-tails ratio. Middle and bottom: Illustration of how different tuned prefixes manipulate the pretrained Transformer's (middle) and LSTM's (bottom) internal state and affect subsequent dynamics. Colored lines are from target distribution (Two-Coin Mixture), gray lines are from pretraining distribution (uniform random coins), same setting as in Fig. A6. Red circles mark the end of the prefixes.

## G.3 Untrained network, fine-tuned on Two-Coin Mixture

Full details on the regret curves and tuning loss curves are given in Fig. A8.

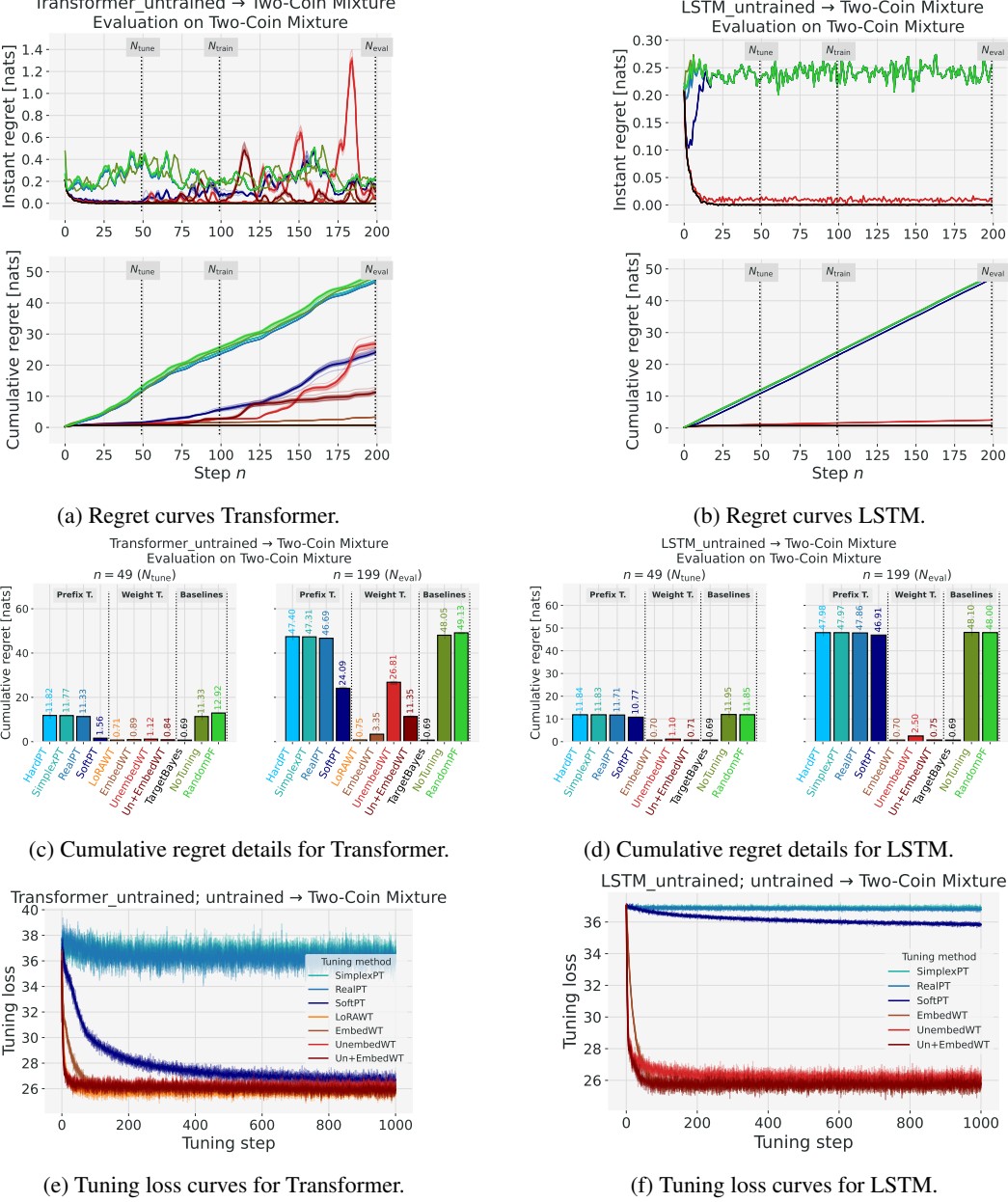

(a) Regret curves Transformer.

(b) Regret curves LSTM.

(c) Cumulative regret details for Transformer.

(d) Cumulative regret details for LSTM.

(e) Tuning loss curves for Transformer.

(f) Tuning loss curves for LSTM.

Figure A8: Untrained models are tuned to a Two-Coin Mixture. Transformer shown in the left column, LSTM shown in the right column. Soft prompting ('SoftPT') a Transformer gets surprisingly close to optimal performance (shown as 'TargetBayes'). Though with poor generalization beyond the tuning sequence length (50 steps). weight-tuning methods perform better, particularly on the LSTM. Tuning loss curves show that 'SoftPT' on the LSTM converges very slowly and may not have fully converged (though it is unlikely that longer training causes a qualitative difference).

## G.4 Untrained network, fine-tuned on Random Coins

Full details on the regret curves and tuning loss curves are given in Fig. A9.

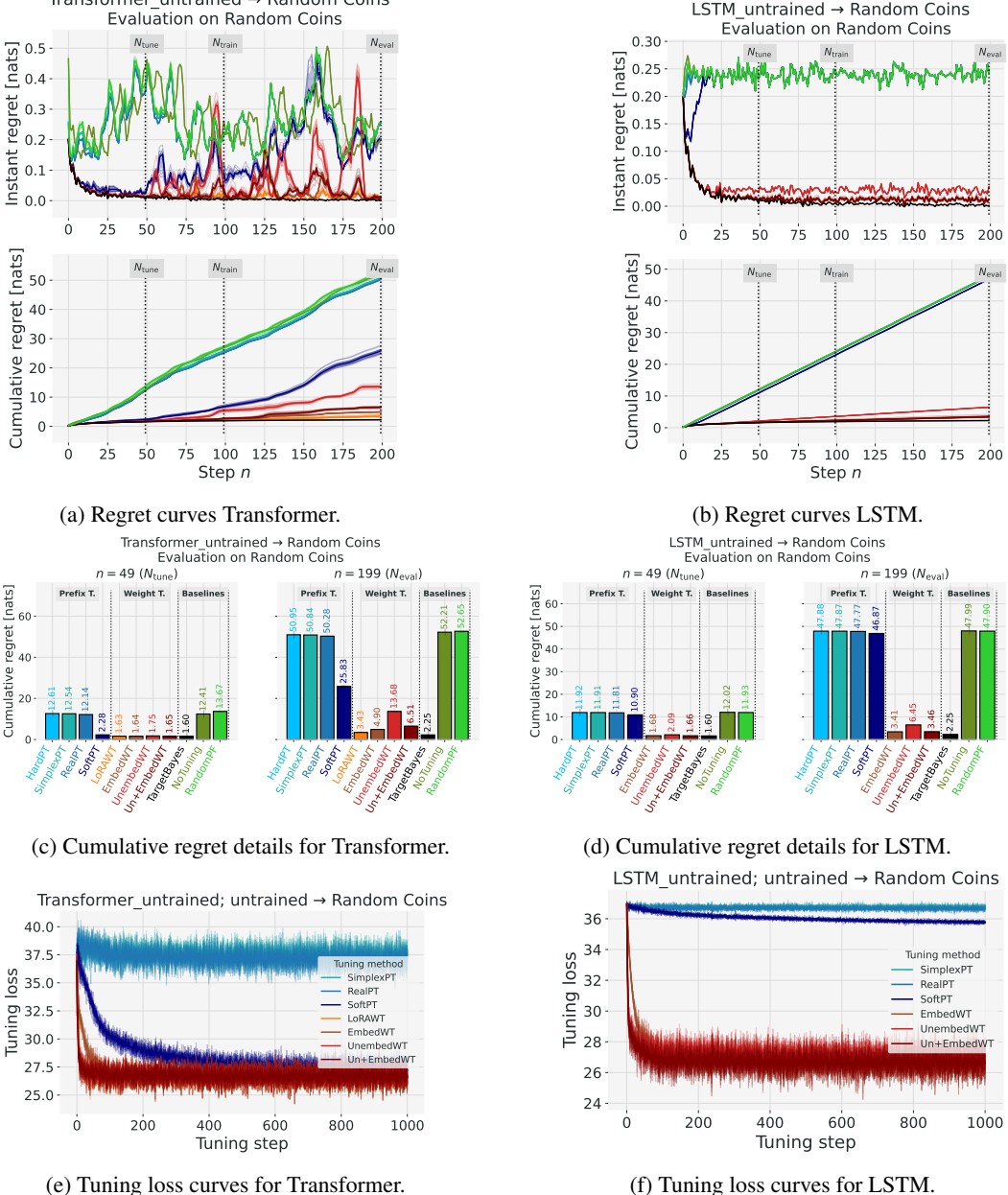

(a) Regret curves Transformer.

(b) Regret curves LSTM.

(c) Cumulative regret details for Transformer.

(d) Cumulative regret details for LSTM.

(e) Tuning loss curves for Transformer.

(f) Tuning loss curves for LSTM.

Figure A9: Untrained models are tuned to (uniform) Random Coins. Transformer shown in the left column, LSTM shown in the right column. Soft prompting ('SoftPT') a Transformer gets close to optimal performance (shown as 'TargetBayes', which is a Laplace predictor in this case and arguably a non-trivial predictor). Though with poor generalization beyond the tuning sequence length (50 steps). Weight-tuning methods perform better, particularly on the LSTM.

# H  prefix-tuning with prefix length 25

Our main experiments use relatively short prefixes of length $L = 6$. The main point is to demonstrate how powerful even very short soft prefixes can be. Additionally, the number of possible hard prefixes grows exponentially with $L$, making exhaustive hard token search for long prefixes intractable. To confirm that our negative result on tuning the pretrained predictor to a Two-Coin Mixture in Fig. 3 is not an artifact of insufficient prefix length, we repeat the soft prefix-tuning experiments with more than triple the prefix length of $L = 25$. Results are shown in Fig. A10. We also show in the same figure that increasing the soft prefix length for tuning untrained networks has only marginal effect.

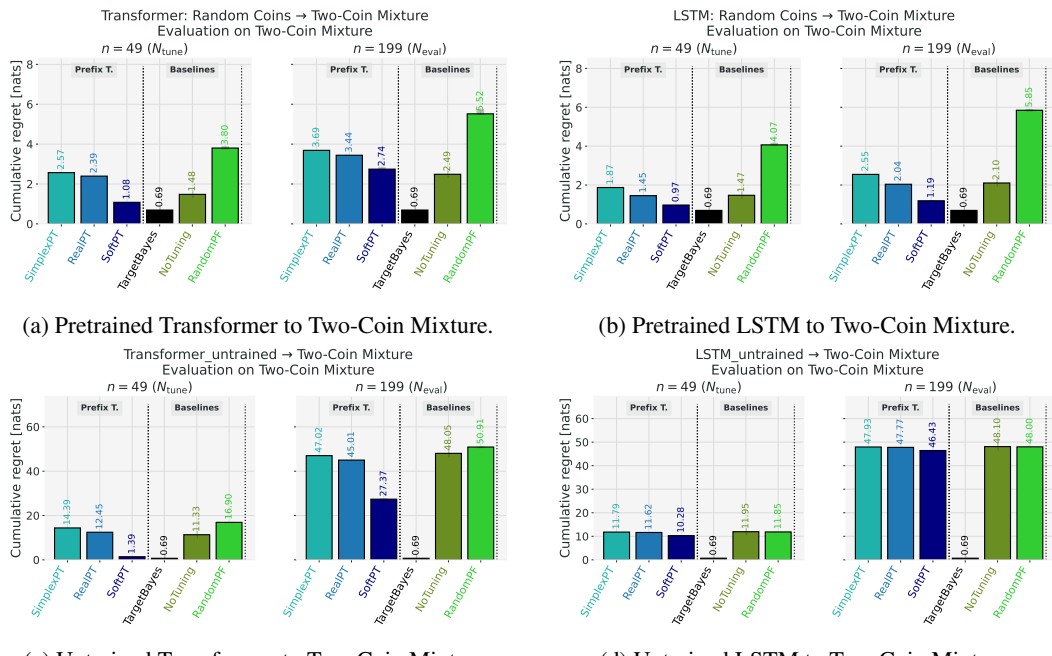

(a) Pretrained Transformer to Two-Coin Mixture.

(b) Pretrained LSTM to Two-Coin Mixture.

(c) Untrained Transformer to Two-Coin Mixture.

(d) Untrained LSTM to Two-Coin Mixture.

Figure A10: prefix-tuning to Two-Coin Mixture with prefix length $L = 25$ (in contrast to all other experiments where $L = 6$). Compare pretrained results with Fig. 3, and untrained results with Fig. 4. Despite a more than tripling the prefix length, only marginal increases in 'SoftPF' performance can be seen in some cases. It is not enough to reach Byaes-optimality on the target distribution, which means that our qualitative results hold.

# I  Reducing the embedding dimensionality

Results shown in Fig. A11 reveal that the superiority of 'SoftPT' over the other soft prefix tuning methods is largely explained by the much higher dimensionality of the embedding vectors (128-dimensional in the main experiments; now reduced to 4 dimensions), compared to input vectors (which are two-dimensional). In frontier models, the input dimensionality is typically higher than the embedding dimensionality, which could in principle result in soft input prefix tuning methods like 'RealPT' outperforming embedding tuning. We leave the question of which prefix tuning method works best at frontier model scale to the large and active research community. Note that since the "width" of our Transformers is equal to the embedding dimensionality (except the width of the MLP layer inside the attention block), the Transformer in our reduced embedding dimensionality experiments is much smaller compared to the main experiments, whereas the LSTM has the same size after the embedding layer.

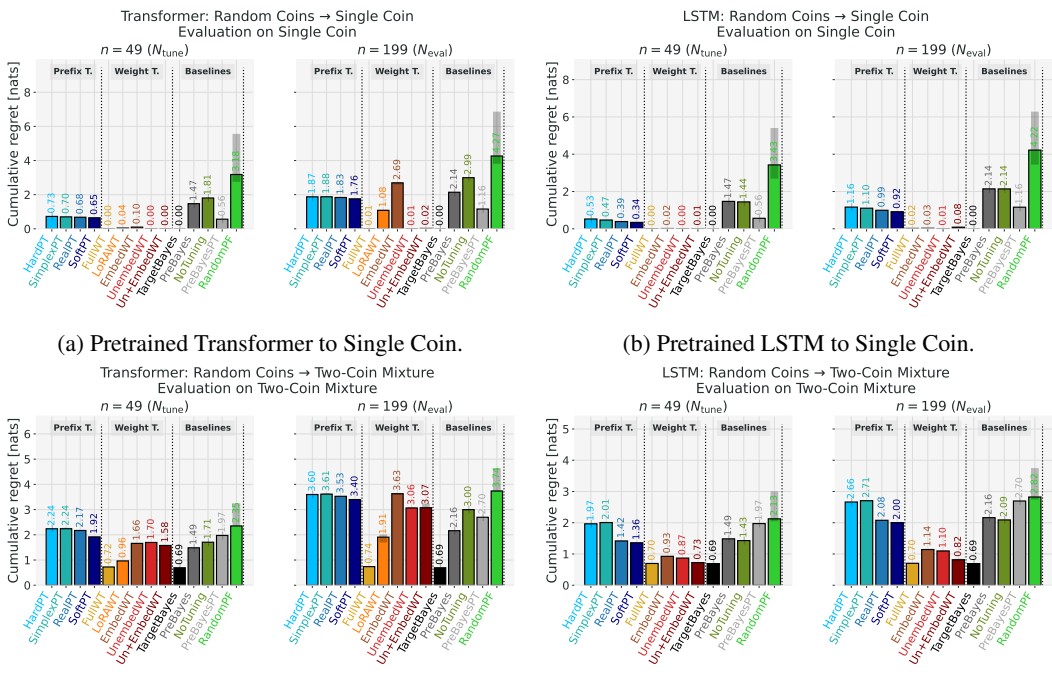

(a) Pretrained Transformer to Single Coin.

(b) Pretrained LSTM to Single Coin.

(c) Pretrained Transformer to Two-Coin Mixture.

(d) Pretrained LSTM to Two-Coin Mixture.

Figure A11: Reducing the embedding dimensionality to 4 (compared to 128 in main experiments) largely eliminates the superiority of 'SoftPT' compared to 'RealPT' (and shortens the gap to the other prefix tuning methods), revealing that the superior performance observed in the main experiments is largely explained by the many more degrees of freedom when tuning soft embedding prefixes vs. soft input prefixes. Compare the results shown here (particularly the dark blue 'SoftPT' bar) against Fig. 1 and Fig. 4 in the main paper. Plots show median results over 3 repetitions.

# J   Experiments with larger networks

Fig. A12 shows results for increasing the network size. Compared to the main experiments we double the embedding dimensionality ($128 \rightarrow 256$), the width of layers ($128 \rightarrow 256$), and the number of layers ($1 \rightarrow 2$). Qualitatively, our main claims hold. Particularly, that prefix tuning can be used to optimally adapt the pretrained predictor to the Single Coin target distribution, but cannot be used for perfect adaptation to the Two-Coin Mixture task. Anecdotally, we have observed our main results to hold robustly, as long as the network size and number of training and tuning steps is large enough. For too small networks, or networks trained or tuned too little, results become more inconsistent. From a theoretical perspective, too small networks violate the realizability condition, and networks with too little training violate the convergence condition.

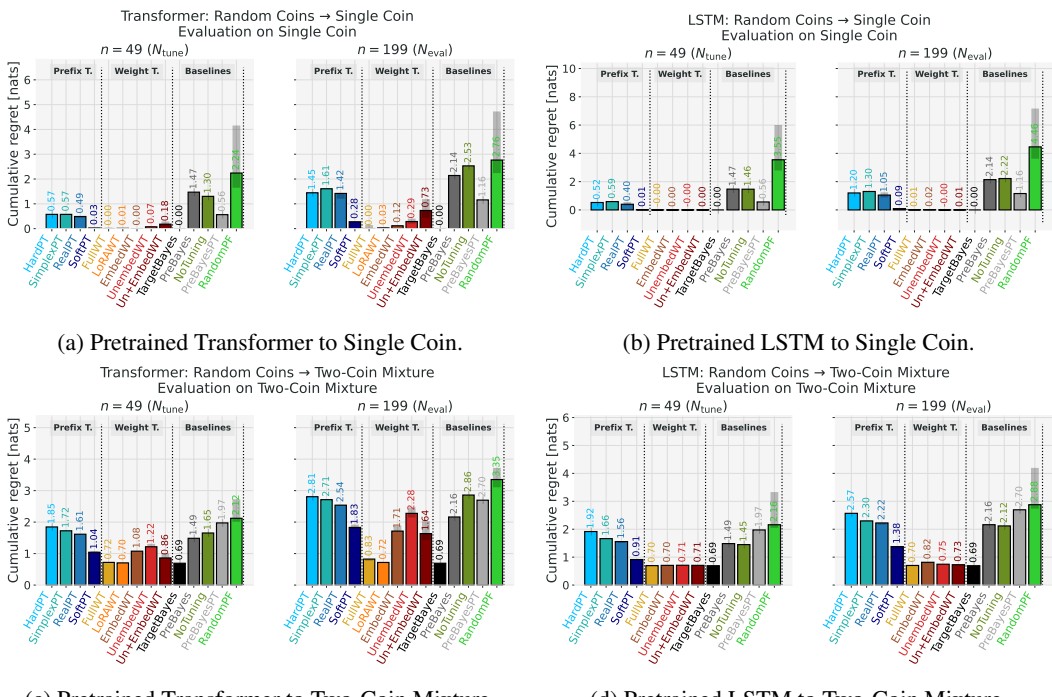

(a) Pretrained Transformer to Single Coin.

(b) Pretrained LSTM to Single Coin.

(c) Pretrained Transformer to Two-Coin Mixture.

(d) Pretrained LSTM to Two-Coin Mixture.

Figure A12: Results for larger networks compared to main experiments are qualitatively equivalent, and show that our main findings are robust against changing model size. Compare the results shown here (particularly the dark blue 'SoftPT' bar) against Fig. 1 and Fig. 4 in the main paper. Plots show median results over 3 repetitions.

