# OpenReview forum: "Understanding Prompt Tuning and In-Context Learning via Meta-Learning"
_NeurIPS.cc/2025/Conference — NeurIPS 2025 spotlight_

### Official Review · Reviewer_PHRz · 2025-06-28

**Clarity:** 4
**Significance:** 3
**Originality:** 3
**Rating:** 5
**Confidence:** 2

**Summary:**

This paper studies prompt tuning under the lens of memory-based meta-learning. It discusses how the problem of finding of optimal prompt for a task can be viewed as conditioning of Bayes predictors. The result of this theory is an understanding of the conditions when optimal prompting is possible or not.

The paper presents a detailed theoretical discussion as well as simulations based on coin-flip data. Interestingly, the paper shows that optimal prompting is not possible when the fine-tuning data and evaluation data is a task mixture.

**Questions:**

I have a few detailed technical questions.

1. Theory for Soft prompting:
 - (1.a) Eq 8 and 9 show when hard prompting is expected to work. However, it does not apply to soft versions, correct?
 - (1.b) If correct, is it possible to extend the theory to deal with SimplexPT, RealPT, SoftPT? I understand these are off-distribution, but is it at least possible to extend the theory for SimplexPT? What is needed? If it is not possible, what exactly is the challenge?
 - Rationale for the question: the empirical results and discussion in Section 5 talks about the superior performance of soft prompts, so it will be nice if this can be captured by the theory.

2. Memory-based meta-learning definition:
 - (2.a) I was not aware of this term. Did it arise from [Ortega 2019] or is there an earlier reference? Also, what exactly is the part that is "memory-based"?
 - (2.b) There have been other papers that propose a Bayesian inference view of in-context learning, such as [Xie 2021] discussed in Section D. How exactly does the memory-based meta-learning framework compare? Is it exactly the same? Is it a generalization? Or are there some nuanced differences despite the similarities?
 - (2.c) Lines 90-95 describes the training process. Are you saying that all LLM pre-training performed by Meta, Google, OpenAi, etc. is essential operating as memory-based meta-learning, except that the task sampling (step 2) is not explicit?
 - (2.d) The Remark in Section 2 says "The main concerns w.r.t. the applicability of the Bayesian viewpoint is that due to limited expressivity, limited data, suboptimal optimization, and off-distribution inputs, models may not converge to or operate in the Bayesian regime." I am not sure I fully understand the implications of this. Expressivity and data can be scaled up in LLMs and suboptimal optimization can be ameliorated with good methods, so in what way is the Bayesian viewpoint not applicable to LLM training in practice?

3. Fig 2: Please provide more explanation on how to interpret the plot.
 - (3.a) What do I make of the difference between random coins and single coin trajectories?
 - (3.b) Where do you observe the off-distribution nature of soft-prefixes?
 - (3.c) What do the red points (n=6) mean, and why are there multiple per graph?
 - (3.d) The general shape for random coins differs between transformer and LSTM. Why?

4. TargetBayes, PreBayes, PreBayesPT: I did not follow how these are constructed and how these differ from each other. Can you provide more description or pseudocode for each separately?

----

Minor comments:
- line 87: typo "preforming a forward pass" --> performing
- I do not quite appreciate the use of longer strings for eval. (N_eval = 200). Does it matter to show generalization beyond tuning length in this experiment?

**Ethical Concerns:**

["NO or VERY MINOR ethics concerns only"]

**Final Justification:**

Strong paper: provides interesting theory and is well-written.

**Limitations:**

yes

**Quality:**

4

**Strengths And Weaknesses:**

Strengths:
 - The paper is extremely well-written. I spent several hours re-reading it and learned a tremendous amount of new material. It reminded me the joys of being a reviewer.
 - The paper provides a theoretical framework for thinking about prompting and in-context learning. This is very valuable for improving our understanding.

I do not see any major weaknesses. Perhaps one might be concerned whether the conclusions on simplified experiments can transfer to real-world language models, but I do not think this decreases the value of the theoretical framework.

---

> ### Author Rebuttal · Authors · 2025-07-30
>
> We thank the reviewer for their feedback and suggestions and are very happy to hear how much they enjoyed reading our paper. Since the aim was to explain and show educational experiments, we particularly appreciate that the reviewer “learned a tremendous amount of new material”. Perhaps the main weakness, that all reviewers identified, is that the relevance of our results at frontier model scale (LLMs/VLMs and data such as text, images, or video) is hard to forecast. We have added this to our limitation section and significantly expanded our discussion section to provide additional details.
>
> **Overview of general changes/additions (across all reviewers):**
> * We discuss the limitations and difficulties with extrapolating our findings to LLM-scale, and some ideas how to approach this (we add to the initial limitations section at the end of the intro, and to the discussion section at the end of the paper). This issue was raised by all reviewers in some form, we added it to our response to HkrS, and kindly ask the other reviewers to read our response there.
> * HkrS asked whether our results are brittle w.r.t. model size. We conducted an additional set of the main experiments (Fig. 1 and Fig. 3) with larger models, and findings are qualitatively equivalent. See our response to HkrS for the main numbers (we cannot provide the full plots that we added to the appendix).
> * PHRz asked whether the superiority of Soft Prompting has a theoretical explanation. As speculated in L298-300, it may simply be the much higher dimensionality of embeddings vs. inputs in our case. We ran a control experiment (for Fig. 1 and Fig. 3) where we reduced the embedding dimensionality from 128 to 4, which largely cancels the superiority of ‘SoftPT’. Full plots were added to the appendix, the most important numbers are in our response to PHRz.
>
> ---
>
> **Specific response:**
>
> Weaknesses:
>
>   * [How would conclusions of the paper transfer to real-world experiments?] We have added another explicit limitation (end of the intro), and some discussion of this question to our discussion section (end of the main paper). See our response to HkrS, who raised a similar issue, where we show the full text that we added.
>
> Questions:
>
>   * 1.a) This is correct - for the theoretical Bayes predictor there is only an input alphabet (of hard tokens). Soft tokens (i.e., real-valued inputs outside the alphabet) are outside this theoretical formulation, which makes it hard to make very general theory-based statements..
>   * 1.b) Unfortunately this is not easily possible. Even for SimplexPT, the inputs are not part of the alphabet of the idealized Bayesian predictor. What happens when we feed such inputs into an implementation of such a predictor depends on the implementation is unclear. E.g., we could implement a Beta-Binomial Bayes predictor in Python - what happens when we feed anything other than the tokens for ‘heads’ and ‘tails’ into this predictor is up to the implementation and not the Bayesian theory.
>   * Superiority of soft prompts: in our case, soft embedding prefixes are in a higher dimensional space compared to the input, which may give more degrees of freedom to “move” in off-distribution activation space and affect the internal representation of the predictor’s memory state. We have briefly speculated that this may be the reason for the superiority of ‘SoftPT’ *in our case* (L298-300). We have added a control experiment which confirms our speculation. We have added the following to the appendix and have clarified in our limitations section (end of introduction) why ‘SoftPT’ is superior to other soft prompt tuning methods in our setting.
>
> “Results shown in Fig. A11 reveal that the superiority of 'SoftPT' over the other (soft) prefix tuning methods is largely explained by the much higher dimensionality of the embedding vectors, compared to input vectors (which are two-dimensional). In frontier models, the input dimensionality is typically higher than the embedding dimensionality, which could in principle result in soft input prefix tuning methods like `RealPT' outperforming embedding tuning. We leave the question of which prefix tuning method works best at frontier model scale to the large and active research community.”
>
> Instead of the full Fig. A11 (which looks similar to Fig. 1 and Fig. 3), we provide here the most important information, which is the question how ‘SoftPT’ performance is affected by the change (compare with ‘RealPT’ in Fig. 1 and Fig. 3)
>
> |                 Setting                | Random -> Single | Random -> Mixture |
> |----------------------------|--------------------|----------------------|
> |    Optimal (TargetBayes)   |             0.00            |              0.69              |
> |      LSTM  (main paper)      |             0.01            |              0.98              |
> | Transformer (main paper) |             0.01            |              1.20              |
> |           LSTM (4-dim)           |             0.34            |              1.36              |
> |      Transformer (4-dim)     |             0.65            |              1.92              |
>
>
>
>   * 2a) There are different forms of meta-learning. Besides memory-based (which dates back a few decades see e.g. Learning To Learn by Thrun & Pratt 1998 and earlier work by Schmidhuber). Another common approach is MAML, where meta-training learns a shared parameter vector (a set of neural network weights) across tasks with the goal of being able to rapidly adapt (via SGD on the weights) to any particular task.
>   * 2b) Other papers, including Xie et al., have identified that in-context learning can be well described as the adaptation of a Bayesian predictor but not where this Bayesian predictor comes from in the first place (the connection to meta-learning theory).
>   * 2c) Yes, in principle LLM training can be viewed as implicit meta-learning (the notion of task may be very subtle, see Ortega et al. 2019 who has a discussion of implicit meta-learning). Perhaps more important than calling LLM pretraining implicit meta-learning is what we get out of this viewpoint.
>   * 2d) Strictly speaking, the Bayesian predictor arises at the endpoint of (perfect) meta-training. If the training process does not fully converge to that point, we have an approximate Bayes predictor but the theory allows us to say relatively little beyond that (whereas we can characterize the endpoint very well). Since it is not clear how close LLMs would be to this endpoint, we think it is important to add this disclaimer.
>   * 3. Thanks for flagging this. We have added more explanation to the main text and figure caption based on your questions. First, please see Fig. A7 a,b, which shows that the pretrained networks essentially use the vertical dimension in the plot (the first principal component) to keep track of the timestep of a sequence, and the horizontal dimension corresponds to the heads-to-tails ratio observed on the current sequence.
>   * 3a) The random coins trajectories are the traces of internal states that one gets by feeding in trajectories from the pretraining distribution (i.e., random coins). The colored single coin trajectories are sequences of internal states that one gets from feeding in sequences of the single coin with fixed bias (and prefixed by prompts tuned via the various methods).
>   * 3b) Since the trajectories of, e.g., SoftPT (dark blue lines) behave very differently from what one would get on pretraining sequences (gray lines), we can visually conclude that they are far off distribution for the Transformer (note that this cannot be seen visually for the LSTM, but because this is a 2D projection of a much more high-dimensional activation vector, it cannot be concluded from these plots that we are ‘on distribution’ for the LSTM).
>   * 3c) The red dots indicate the internal state of the neural network after it has consumed the prefix of length 6.
>   * 3d) Some differences in internal structure for different architectures are expected. What is interesting (Fig. A7 a,b) is that for both architectures the same two signals (number of steps and heads-to-tails ratio) dominate the variance of the internal state (the first two principal components). These are exactly the task’s sufficient statistics, and in this abstract sense the structure is very similar between both network types.
>   * 4) See lines 251-253 in the paper. These three predictors are different known (analytical) Bayesian predictors. For pre-training on ‘Random Coins’, PreBayes is a Beta-Binomial distribution. For fine-tuning to a single coin ‘TargetBayes’ is a Binomial distribution with the correct coin bias. For the Two-Coin Mixture ‘TargetBayes’ is a mixture of two Binomial distributions. ‘PreBayesPT’ results from taking the ‘PreBayes’ (Beta-Binomial), and prompting it with an optimal sequence of 6 hard tokens for the target task (which we find via exhaustive search).
>
>
> Minor comments:
>
> Thanks for pointing out the typo. Re longer sequences: the paper and main claims hold when considering results at n=49, which is where we have strict theoretical guarantees. We nonetheless think the additional results are important and interesting to get a sense of how robust the performance is for various tuning methods. E.g., it is conceivable that ‘SoftPT’ would produce very good performance up to the length for which the soft prompts were optimized (i.e. n=49), and then drops sharply because the off-distribution nature of the resulting internal activations rapidly “spirals out of control”, but we find this not to be the case.

---

> > ### Comment · Reviewer_PHRz · 2025-08-05
> >
> > Thanks for the additional answers. Great work!

---

> > > ### Author Response · Authors · 2025-08-07
> > > **Thanks for the review**
> > >
> > > Thanks for having a look at our rebuttal. We are happy to see that the reviewer has no additional questions.
> > >
> > > We noticed that we can still see the reviewer's score, which probably means that the reviewer has not yet entered their final justification (by editing the original review). Just flagging this as the AC will probably sooner or later raise this.

---

### Official Review · Reviewer_QjaG · 2025-07-02

**Clarity:** 3
**Significance:** 2
**Originality:** 4
**Rating:** 5
**Confidence:** 3

**Summary:**

This paper describes prompt tuning through a Bayesian lens and runs a number of experiments with many prompt tuning baselines in a synthetic setting. They include some mechanistic analysis of the proposed methods.

**Questions:**

- Is 1000 steps of pretraining really enough training to learn the distribution of random coin flips? I would expect this to be hard to learn and general require much more training.
- This is a tiny nitpick but it might be cleaner to name the embedding dimension 'd' instead of writing out 'Embedding-dimensionality' in superscript in S3.
- How was HardPT implemented? Did you try a state-of-the-art discrete optimizer such as GCG (https://arxiv.org/abs/2307.15043)?

'The embedding is a trainable linear projection from the 2D token space into a 128-dimensional “embedding” space.'
   Is the input not a single dimension? (1 or 0)

- A small presentation question: What is the main intended takeaway from Figure 1? Clarifying language in the caption may go a long way here.

**Ethical Concerns:**

["NO or VERY MINOR ethics concerns only"]

**Final Justification:**

After reviewing the rebuttal and the discussions between the authors and the other reviewers, I decided to increase my score from 4 (Borderline Accept) to 5 (Accept).

**Limitations:**

yes

**Quality:**

2

**Strengths And Weaknesses:**

Strengthes:
- Prompt/prefix tuning is important, interesting, and understudied.
- The theoretical analysis seems sound. Exploring LLM-related topics through a Bayesian lens seems important.
- The baselines are thorough and include all the prompt tuning methods I would expect (except perhaps some missing variants of HardPT).
- The experiments on untrained transformers surprised me and could translate to text data.

Weaknesses:
- Experiments are limited to randomly generated synthetic datasets. It is not clear how if at all any findings might be useful in real-world scenarios.
- The Bayesian lens doesn't appear to tell us anything new about how prompt tuning works.
- There are many ways to implement "HardPT" and there isn't much if any discussion of this.

---

> ### Author Rebuttal · Authors · 2025-07-30
>
> We thank the reviewer for their feedback and suggestions, and are happy to hear that they found our work very original and clear. Perhaps the main weakness, that all reviewers identified, is that the relevance of our results at frontier model scale (LLMs/VLMs and data such as text, images, or video) is hard to forecast. We have added this to our limitation section and significantly expanded our discussion section to provide additional details.
>
> **Overview of general changes/additions (across all reviewers):**
> * We discuss the limitations and difficulties with extrapolating our findings to LLM-scale, and some ideas how to approach this (we add to the initial limitations section at the end of the intro, and to the discussion section at the end of the paper). This issue was raised by all reviewers in some form, we added it to our response to HkrS, and kindly ask the other reviewers to read our response there.
> * HkrS asked whether our results are brittle w.r.t. model size. We conducted an additional set of the main experiments (Fig. 1 and Fig. 3) with larger models, and findings are qualitatively equivalent. See our response to HkrS for the main numbers (we cannot provide the full plots that we added to the appendix).
> * PHRz asked whether the superiority of Soft Prompting has a theoretical explanation. As speculated in L298-300, it may simply be the much higher dimensionality of embeddings vs. inputs in our case. We ran a control experiment (for Fig. 1 and Fig. 3) where we reduced the embedding dimensionality from 128 to 4, which largely cancels the superiority of ‘SoftPT’. Full plots were added to the appendix, the most important numbers are in our response to PHRz.
>
> ---
>
> **Specific response:**
>
>
> Weaknesses:
>
>   * [Experiments only on synthetic datasets.] We have added this explicitly to the limitations section at the end of the introduction and to our discussion section at the end of the paper. Please see our response to HkrS for the precise text that we added.
>   * [The Bayesian lens does not tell anything new about how prompt-tuning works.] We respectfully disagree. Our theoretical analysis, centered on the Bayesian view, reveals novel limitations of when prefix-tuning can and cannot work regardless of the tuning method, which we confirm to hold in simple experiments, but the theory holds at any scale (incl. LLM scale and beyond). We are not aware of any previously published work predicting or providing an explanation of why prompt-tuning to any single task works but prompt-tuning to any mixture of two or more tasks is at best suboptimal. Additionally, the Bayesian view emphasizes that prompting works via manipulating the sufficient statistics of a Bayesian predictor (in practice in ways that go beyond strict statistical conditioning, e.g., when using real-valued prefixes), which we have not seen spelled out in the literature before, and which we illustrate by recording how the internal state of networks is manipulated by various kinds of prompts. We thus used the Bayesian lens to tell us quite a bit new about how prompt-tuning works on a very fundamental level, and to a smaller degree mechanistically.
>   * [There are many ways to implement HardPT.] This is correct, and in general hard token tuning is a challenging problem with many published methods. We avoid this problem in our study by performing “exhaustive search” (L138, L260 in the paper). This means that we evaluate model performance for *every possible* hard-token prefix (2^6=64 possible hard prefixes in our case). Accordingly, the prefix we find cannot be improved with any other method of hard-token tuning.
>
> Questions:
>   * [Is 1000 steps of pretraining enough?] Yes. We investigate pretraining curves for all our experiments (and they can be easily produced with the released code), and the 1000 steps were chosen via pilot experiments to suffice for convergence in all experiments with a comfortable margin.
>   * [Rename ‘Embedding-dimensionality’.] We agree the notation is a bit awkward, but it has the advantage that it does not need any further explanation / symbol. Since we only use it once in the paper, we have left it as is for now, but will change it if the reviewer feels strongly about it.
>   * [How was HardPT implemented?] As stated above (and L138 in the paper) via exhaustive search over all 64 possible binary hard token sequences of length 6.
>   * [Input dimensionality vs. embedding dimensionality.] Yes, the input dimensionality is 2 (binary inputs, i.e. two-dimensional one-hot encodings of ‘0’ and ‘1’). We pass inputs through a first linear projection (with learnable parameters) to compute an “embedding”.
>   * [What is the main takeaway from Fig. 1?] The main takeaway from Fig. 1 is that the pretrained network *can be successfully prompted to behave like the Bayes-optimal predictor on the target distribution (TargetBayes) from the very beginning of eval sequences onwards*. The effect of optimal prompting thus becomes very similar to weight-tuning on the target task *in this case*. We have added this main takeaway to the figure caption.

---

### Official Review · Reviewer_b8X5 · 2025-07-03

**Clarity:** 3
**Significance:** 2
**Originality:** 3
**Rating:** 5
**Confidence:** 3

**Summary:**

This paper provides a conceptual framework of viewing the large pretrained LLMs as meta-trained neural networks which can behave as Bayesian predictors over their pretraining distributions. It reinterprets prompt tuning as conditioning those Bayesian predictors with short prefixes. It proves that prompting can reach bayes-optimal performance only when the target task already lies inside the pretraining support and is unimodal. It shows two formal failure modes: multimodal targets and tasks never seen in pretraining. It validates the theory with toy coin-flip tasks using tiny ttransformers and LSTMs. It finds soft prompts inject more information than any hard token sequence. It also shows weight tuning sidesteps the theoretical limits but overwrites the original model. The paper is overall clearly written and insightful yet confined to trivial data, small models.

**Questions:**

1. The educational experiments are only conduced on small models like LSTMs. How do you predict your results might translate to these more modern LLMs like GPT series, LLama.
2. How your bayesian optimization and meta learning view explains the in-context learning ability's brittleness in context order, prompt formats sensitivities?
3. What actionable guidance does your analysis offer for practitioners—e.g., on choosing prefix length, deciding between soft-prompt tuning and weight-tuning?

**Ethical Concerns:**

["NO or VERY MINOR ethics concerns only"]

**Final Justification:**

Their rebuttal provided new experiments and I think their theory is clean, and have some insights for practitioners as well. Therefore raising score to 5.

**Limitations:**

yes

**Quality:**

3

**Strengths And Weaknesses:**

Strengths:
1. This paper introduces a novel view of viewing the large-corpus pretraining of LLM as meta-training, where each training example is conditioned on an task and then the LLM becomes a sequential bayesian predictor, And this paper recasts prompting as steering this bayesian predictor.
2. it has analyzed theoretical conditions for which the optimal prompting is and is not possible. And they empirically demonstrated this claim on atomic tasks.
3. Through educational expeirments, they comfirm their theoretical conditions results.
4. it also includes visual evidence in the PCA plots which shows how different prefixes physically move the hidden states into the region which means a good posterior region.

Cons:
1. The educational experiments might not extend to real-world tasks where the combinatorial structure of language, vision or other modalities.
2. the experiments are only conduced on small transformers and LSTM.

---

> ### Author Rebuttal · Authors · 2025-07-30
>
> We thank the reviewer for their feedback and suggestions, and are happy to hear that they enjoyed reading our paper and rated it as good on 3 out of 4 dimensions. Perhaps the main weakness, that all reviewers identified, is that the relevance of our results at frontier model scale (LLMs/VLMs and data such as text, images, or video) is hard to forecast. We have added this to our limitation section and significantly expanded our discussion section to provide additional details.
>
> **Overview of general changes/additions (across all reviewers):**
> * We discuss the limitations and difficulties with extrapolating our findings to LLM-scale, and some ideas how to approach this (we add to the initial limitations section at the end of the intro, and to the discussion section at the end of the paper). This issue was raised by all reviewers in some form, we added it to our response to HkrS, and kindly ask the other reviewers to read our response there.
> * HkrS asked whether our results are brittle w.r.t. model size. We conducted an additional set of the main experiments (Fig. 1 and Fig. 3) with larger models, and findings are qualitatively equivalent. See our response to HkrS for the main numbers (we cannot provide the full plots that we added to the appendix).
> * PHRz asked whether the superiority of Soft Prompting has a theoretical explanation. As speculated in L298-300, it may simply be the much higher dimensionality of embeddings vs. inputs in our case. We ran a control experiment (for Fig. 1 and Fig. 3) where we reduced the embedding dimensionality from 128 to 4, which largely cancels the superiority of ‘SoftPT’. Full plots were added to the appendix, the most important numbers are in our response to PHRz.
>
> ---
>
> **Specific response:**
>
>
> Weaknesses:
>
>   * [Educational experiments might not extend to real-world tasks]. We have added this as an explicit limitation and have expanded our discussion. Please see our response to HkrS for details.
>   * [Experiments are only conducted on small networks]. We have included an additional experiment in the appendix with twice the embedding dimensionality, twice the layer width and double the number of layers (two instead of one). Qualitative results are equivalent and all our main claims hold. The experiments mainly address HkrS’s question about how robust our results are to model scale. Please find more details in our response to HkrS. We do acknowledge that compared to frontier model scale, these networks are still “small”.
>
> Questions:
>   * [How do we predict our experiments would translate to frontier models?] We have added this as an explicit limitation and have expanded our discussion. Please see our response to HkrS.
>   * [What actionable guidance do we have for practitioners?] Our original discussion section was meant to address this question, but rather than speculating we have formulated a number of hypotheses to investigate in the discussion. In particular:
>      * Soft prefix tuning (SoftPT, RealPT, and modern methods based on it) should be superior to hard token tuning and traditional prompt engineering.
>     * This may be particularly fruitful for in-context imitation learning, where one typically has small datasets available on which soft prefixes could be tuned, but the standard practice is to simply add a number of demonstrations to the context.
>     * One take-away for practitioners based on our experiments could be to always use weight tuning (full weights or LoRA), as that yields the same or better performance as prefix tuning and does not suffer from its theoretical limitations. However, as we noted in the discussion this may be a wrong conclusion, as the two methods come with different practical implications and other works (e.g., Lampinen et al. 2025, Chan et al. 2022b) have found diminished neural plasticity after LLM pretraining, making weight tuning potentially less effective compared to prompt tuning.
>     * Generally, we advise practitioners to compare prefix tuning *and* weight tuning, whenever possible, and pick which works best.
>   * [How does the Byaesian view on in-context learning explain the brittleness and sensitivity to non-semantic changes to prompts?] In short: these issues are outside our theory and there is not much that can be said about them from this particular angle (without significantly extending the theory). This does not invalidate the theory and our findings, it simply says that for certain (important) questions, the theory is not enough / not appropriate. For instance, we observed in previous work that network predictions and internal states on a coin-flip task are not fully invariant to the order with which samples are shown (an ideal Bayesian predictor only cares about counts of heads and tails, and is invariant to the order). This is not completely surprising, as learning perfect invariances over a large space of possible strings would be very difficult. So in practice, meta-trained neural nets can deviate from the ideal Bayesian predictor, and in these cases the idealized Bayesian theory cannot be used to make precise predictions about some very particular questions. Nonetheless, and despite these issues, our experiments show that the Bayesian theory is sharp and relevant on an abstract level, and theory tells us that, in principle, as models become even more expressive and optimizers get better, the theory becomes more relevant and accurate.

---

> > ### Comment · Reviewer_b8X5 · 2025-08-07
> >
> > Thank your for the responses and the new experiments.  The partitioners guidance is also helpful. I have raised my score.

---

### Official Review · Reviewer_HkrS · 2025-07-16

**Clarity:** 3
**Significance:** 4
**Originality:** 4
**Rating:** 5
**Confidence:** 4

**Summary:**

This paper shows that meta-trained neural networks can be approximated as Bayes-optimal predictors and explore which prefix tuning technique results in optimal prompting (Bayes-optimal). To investigate this, the paper first conducts theoretical analysis, where it derives a connection between minimizing expected log loss over a distribution of stochastic data generators and the Bayes-optimal predictor for that pre-training distribution. For empirical analysis, the authors used an educational experimental setup using coin-flip sequences, where the exact Bayes-optimal predictor is analytically known. Also they train a basic Transformer and a LSTM architecture model, comparing the performance of various prefix tuning strategies against traditional weight tuning and other baselines. The empirical findings indicate that prefix tuning methods do not reach Bayes-optimal. Weight tuning methods achieve better results than prefix-tuning, although there is still a small gap when compared against Bayes-optimal.

**Questions:**

NA

**Ethical Concerns:**

["NO or VERY MINOR ethics concerns only"]

**Final Justification:**

The authors have provided additional results and details on what is the limitation. Overall a good contribution. I have revised the score based on the responses.

**Limitations:**

Yes

**Paper Formatting Concerns:**

Line 129,  ‘alphabet’ -> “alphabet”

Line 240, ‘error bars’, line 252 and 256 also has text with single quote, make them double quote unless they were single quotes intentionally. There are few more places with similar single quote.

Line 242 baselienes, misspelled

**Quality:**

4

**Strengths And Weaknesses:**

Strengths

- I really like the idea of connecting meta-learning to Bayesian predictors.
- The theoretical analysis is well-grounded.
-  Clean and controlled experimental setup using coin-flip sequences, where the Bayes-optimal predictor is analytically known. This allows for precise evaluation of different tuning methods.
- The authors also compare an exhaustive set of techniques, including prefix tuning variants and full weight tuning, providing a comprehensive view of their relative performance.

Weaknesses

- Even though the coin flip provides a good starting point, it does not capture the complexity and variability of real-world data. Therefore how to extend the study to complex real world tasks remains unclear. Extending the study to include richer synthetic tasks or controlled natural datasets (e.g., language modeling on short-form text, or structured reasoning problems) would provide stronger evidence for the claims.
- As it might be difficult to get Target Bayes for real world tasks, is there a way to approximate this optimal value? Which will help point 1 too.
- Is the result pattern we see on the coin-flip dataset consistent with another model size or is specific to the chosen model? A few selected experiments may help answer this.

---

> ### Author Rebuttal · Authors · 2025-07-30
>
> We thank the reviewer for their feedback and suggestions, and are happy to hear that they enjoyed reading our paper and rated it as excellent on 3 out of 4 dimensions. Perhaps the main weakness, that all reviewers identified, is that the relevance of our results at frontier model scale (LLMs/VLMs and data such as text, images, or video) is hard to forecast. We have added this to our limitation section and significantly expanded our discussion section to provide additional details.
>
> **Overview of general changes/additions (across all reviewers):**
> * We discuss the limitations and difficulties with extrapolating our findings to LLM-scale, and some ideas how to approach this (we add to the initial limitations section at the end of the intro, and to the discussion section at the end of the paper). This issue was raised by all reviewers in some form, we added it to our response to HkrS, and kindly ask the other reviewers to read our response there.
> * HkrS asked whether our results are brittle w.r.t. model size. We conducted an additional set of the main experiments (Fig. 1 and Fig. 3) with larger models, and findings are qualitatively equivalent. See our response to HkrS for the main numbers (we cannot provide the full plots that we added to the appendix).
> * PHRz asked whether the superiority of Soft Prompting has a theoretical explanation. As speculated in L298-300, it may simply be the much higher dimensionality of embeddings vs. inputs in our case. We ran a control experiment (for Fig. 1 and Fig. 3) where we reduced the embedding dimensionality from 128 to 4, which largely cancels the superiority of ‘SoftPT’. Full plots were added to the appendix, the most important numbers are in our response to PHRz.
>
> ---
>
> **Specific response:**
>
> Weaknesses:
>
>   * [Synthetic data too simplistic; models too small; how to extend the study to real world tasks]. We have made the following additions to the paper.
>
> Addition to limitations section (end of intro):
> “This paper discusses fundamental properties of prefix tuning, which we illustrate with educational experiments where the focus is on clarity and being able to compare against a tractable Bayesian predictor. Accordingly, our datasets do not capture the full complexities of, e.g, large-scale language, vision, or robotics tasks. Similarly, our neural networks are small by modern standards, which means that our findings must be very cautiously extrapolated towards modern frontier model scale. Further rigorous and well-designed scientific studies are necessary to bridge the gap between our current work and modern large-scale ML practice, and we are optimistic that our fundamental results will inspire the design of such studies.”
>
> Addition to the beginning of the discussion section:
> “While we have laid important fundamental groundwork in our current study, extrapolating our findings to modern frontier model (and data) scale is not straightforward. While the theoretical findings we presented, including the limitations of prefix tuning, hold at any scale, it is likely that additional practical issues arise at scale that are not captured by our current small-scale experiments. It is thus hard to predict the relevance and impact of our fundamental results on today’s frontier-model practice. For instance, one of our main results is that optimal prompting to a single target task is possible, whereas it is not for a mixture of tasks. We are confident that this holds even at frontier model scale (based on the theory), but it is unclear and highly non-trivial what constitutes a task for a LLM, and accordingly, whether this is a severe limitation or not. In our experiments, a task is simply an unobserved variable in a two-level hierarchical statistical model. At LLM scale, the structure is vastly more complex, with many more hierarchical levels, and potentially other statistical structures at play. The next step would be to carefully design data generators that are closer to natural language data, but still fully understood and well controllable, akin to the data generators used, e.g., in the `Physics of LLMs’ paper [1], and run our experiments of tuning to single tasks vs. tuning to mixtures of tasks at scale.
> With these caveats in mind, cautiously extrapolating our findings to frontier model scale, raises some questions for investigation [the orig. discussion section then follows...]”
>
>   * [How could we get TargetBayes for real world tasks?]. For many real-world tasks, all kinds of data modeling techniques are used to approximate TargetBayes. As models and optimizers get better, these approximations will get better. Note though, that we do not necessarily need TargetBayes: different prompt- or weight-tuning methods can be meaningfully compared against each other w.r.t. their cumulative log loss on a held out evaluation set. It would just be unclear how far these methods are from the optimum.
>   * [Is the result pattern consistent with other model sizes?]. We have observed the qualitative pattern underlying our main results very robustly across pilot experiments with different model sizes. To be rigorous, we have conducted an experiment with larger models, and have added the following to the appendix:
>
> “Fig. A12 shows results for increasing the network size. Compared to the main experiments we double the embedding dimensionality (128->256), the width of layers (128->256), and the number of layers (1->2) [called LSTM Maxi and Transformer Maxi in the table below]. Qualitatively, our main claims hold. Particularly, that prefix tuning can be used to optimally adapt the pretrained predictor to the Single Coin target distribution, but cannot be used for perfect adaptation to the Two-Coin Mixture task. Anecdotally, we have observed our main results to hold robustly, as long as the network size and number of training and tuning steps is large enough. For too small networks, or networks trained or tuned too little, results become more inconsistent. From a theoretical perspective, too small networks violate the realizability condition, and networks with too little training violate the convergence condition. The theory thus does not apply to predict outcomes in this regime.”
>
> Instead of the full Fig. A12 (which looks similar to Fig. 1 and Fig. 3), we only provide the most important information in the rebuttal: whether ‘SoftPT’ can be used to reach optimality or not?  The answer is true for ‘Random -> Single’ but not for ‘Random -> Mixture’.
>
> |                 Setting                | Random -> Single | Random -> Mixture |
> |----------------------------|--------------------|----------------------|
> |    Optimal (TargetBayes)   |             0.00            |              0.69              |
> |      LSTM  (main paper)      |             0.01            |              0.98              |
> | Transformer (main paper) |             0.01            |              1.20              |
> |               LSTM Maxi            |             0.01            |              0.91              |
> |         Transformer Maxi       |             0.03            |              1.04              |
>
>
> * Formatting issues: thanks for pointing out these typos. We have fixed them in our updated manuscript.
>
> [1] Allen-Zhu, Zeyuan, and Yuanzhi Li. "Physics of language models: Part 3.1, knowledge storage and extraction." arXiv preprint arXiv:2309.14316 (2023).

---

> > ### Comment · Reviewer_HkrS · 2025-08-06
> > **Thanks for the responses.**
> >
> > Thanks for additional result and clarifying the limitation in more details. I have revised my score. Thanks

---

> > > ### Author Response · Authors · 2025-08-07
> > > **Thanks for the review**
> > >
> > > Thanks for considering our rebuttal. We are pleased to hear that we could address the reviewer's criticism and questions to a sufficient degree.

---

### Note · Authors · 2025-08-12

We thank the reviewers for their effort and engagement, and given the responses we have **nothing further to add** at this point. For an overview of all changes/additions see the top of any of our responses to a particular reviewer.

---

### Decision · Program_Chairs · 2025-09-17

**Decision:**

Accept (spotlight)

**Comment:**

This is an overall interesting paper that is well executed. The reviewers are all positive, mention that their concerns are all addressed and are generally happy with the paper. There are some nice foundational contributions into understanding various forms of adaptation. The didactic experiments are nice and well-explained. The limitations (acknowledged by the authors) are that it's not clear how to extrapolate results to frontier models. What keeps me from giving an even higher score is that the findings are more-or-less already known and not super surprising, but it is still very valuable to put all these "folklore" knowledge on a precise footing with careful experiments. I recommend acceptance.